# Tropospheric sulfate from Cumbre Vieja (La Palma) observed over Cabo Verde contrasted to background conditions - lidar case study of aerosol extinction, backscatter, depolarization and lidar ratio profiles at 355, 532 and 1064 nm

Henriette Gebauer[1,2], Athena Augusta Floutsi[1], Moritz Haarig[1], Martin Radenz[1], Ronny Engelmann[1], Dietrich Althausen[1], Annett Skupin[1], Albert Ansmann[1], Cordula Zenk[3,4], and Holger Baars[1]

[1]Leibniz Institute for Tropospheric Research, Leipzig, Germany
[2]Institute for Meteorology, Leipzig University, Leipzig, Germany
[3]Ocean Science Center Mindelo, Mindelo, Cabo Verde
[4]GEOMAR Helmholtz Centre for Ocean Research, Kiel, Germany

**Correspondence:** Henriette Gebauer (gebauer@tropos.de)

**Abstract.** In September 2021, volcanic aerosol (mainly freshly formed sulfate plumes) originating from the eruption of Cumbre Vieja on La Palma, Canary Islands, Spain, crossed Cabo Verde at altitudes below 2 km. On 24 September 2021, an extraordinarily large aerosol optical depth (AOD) close to 1.0 (daily mean at 500 nm) was observed at Mindelo, Cabo Verde. This event provided favorable conditions to obtain lidar-derived profiles of extinction and backscatter coefficients, lidar ratio and depolar-

ization ratio at 355, 532 and 1064 nm in the sulfate aerosol plume. A novel feature of the lidar system operated at Mindelo is the availability of extinction, lidar ratio and depolarization measurements at 1064 nm in addition to the standard wavelengths of 355 and 532 nm. Having measurements of these parameters at all three wavelengths is a major advantage for the aerosol characterization and in aerosol typing efforts as the lidar ratio and the particle linear depolarization ratio are key parameters for this purpose. In this article, we present the key results of the lidar observations obtained on one specific day, namely on

24 September 2021, 04:38–05:57 UTC, including the first ever measurements of the particle extinction coefficient, the lidar ratio and the depolarization ratio at 1064 nm for volcanic sulfate, and discuss the findings in terms of aerosol optical properties and mass concentrations by comparison to a reference observation (16 September 2021) representing the typical background conditions before the start of the eruptions. We found an unusual high particle extinction coefficient of $721 \pm 51$, $549 \pm 38$ and $178 \pm 13\,\mathrm{Mm^{-1}}$ and an enhanced lidar ratio of $66.9 \pm 10.1$, $60.2 \pm 9.2$ and $30.8 \pm 8.7\,\mathrm{sr}$ at 355, 532 and 1064 nm, respectively,

in the sulfate-dominated planetary boundary layer (PBL). The particle linear depolarization ratio was $\leq 0.9\,\%$ at all respective wavelengths. It is the first time that lidar-derived intensive aerosol optical properties could be derived for volcanic sulfate at all three wavelengths and, thus, it is a highly valuable data set for global aerosol characterization. The lidar analysis also revealed a sulfate-related AOD of about $0.35 \pm 0.03$ at 532 nm of the total PBL-related AOD of 0.43. The rest of the AOD contribution was caused by a lofted Saharan dust layer extending from 1.4 to 5 km and leading to a total AOD of 0.79 at 532 nm. Volcanic

ash contribution to the observed aerosol plumes could be mostly excluded based on trajectory analysis and the observed op-

tical properties. Peak mass concentration was $178.5 \pm 44.6 \, \mu gm^{-3}$ in the volcanic influenced – sulfate dominated polluted PBL showing the hazardous potential of such sulfate plumes to significantly worsen local air quality at even remote locations.

# 1 Introduction

Volcanic eruptions are of large importance for the Earth's climate (Hansen et al., 1997; Robock, 2000) because the emitted
particles and gases can be transported several hundreds of kilometers away from the source and influence the global radiation budget (Solomon et al., 2011; Groß et al., 2012; Martin et al., 2014). Typically emitted products of volcanic activity are ash particles with a diameter smaller than 2 mm during explosive phases, as well as volatiles such as sulfur dioxide ($SO_2$; McGonigle et al., 2004; Aiuppa et al., 2008; Carracedo et al., 2022). $SO_2$ is the most abundant gas emitted by volcanoes (Kampouri et al., 2021) of which 10–20 Mt are released into the troposphere each year (Martin et al., 2014). While in the
stratosphere this gas has a lifetime of multiple weeks, it persists in the troposphere for around 1–3 days (Navas-Guzmán et al., 2013; Pattantyus et al., 2018). In a chemical reaction with water and further atmospheric components (hydroxyl radical (OH) in clear air conditions or hydrogen peroxide ($H_2O_2$) in cloudy air), it is quickly converted to sulfate aerosol ($SO_4^{2-}$ bearing substances and sulfuric acid droplets; Ansmann et al., 2011b; Martin et al., 2014; Pattantyus et al., 2018). The efficiency of the conversion of $SO_2$ to sulfate aerosol is influenced by multiple factors and increases with temperature and relative humidity
(Eatough et al., 1994; Yang et al., 2018). The lifetime of sulfate aerosol in the troposphere of 1 to 3 weeks is much longer than the one of $SO_2$ or volcanic ash so that it can be transported over long distances (Pappalardo et al., 2004; Filonchyk et al., 2022). If it reaches the higher troposphere/lower stratosphere it can remain even for several years (Jäger, 2005; Deshler, 2008; Martin et al., 2014).

   Sulfate aerosol particles are impacting the climate in several ways since they reflect solar radiation (Pappalardo et al., 2004)
and scatter light even more efficiently with increasing relative humidity due to hygroscopic growth (Miffre et al., 2012). Furthermore, they act as cloud condensation nuclei (CCN) and ice nucleation particles (INPs) and, thus, influence the precipitation cycle (Pappalardo et al., 2004). Especially in cities, sulfate aerosol is of large importance with regard to air quality. It is one of the major components of urban PM2.5 (Zhang et al., 2004; Yang et al., 2018). Although anthropogenic $SO_2$ emissions are with 110 Mt per year 5–10 times higher than volcanic emissions, volcanic eruptions are one of the greatest natural sources for sulfur
emissions (Martin et al., 2014). Furthermore, their emissions have a larger impact on the climate due to the release of $SO_2$ in higher altitudes, which provides a longer lifetime of the formed aerosol particles (Kampouri et al., 2021). In addition, sulfate particles can be emitted directly as well (Martin et al., 2014). Moreover, volcanic eruptions have not only climatological but also economical consequences with regard to aviation. For example, volcanic ash can cause engine damage at aircraft or the air traffic is even suspended, as happened during the eruption of the Iceland volcano Eyjafjallajökull in spring 2010 (Groß et al.,
2012). To reduce the risks, ash-dispersion simulations are used in early warning systems. Assimilation of satellite products like Aeolus wind measurements improve the ash plume forecast, as shown in a recent study of Amiridis et al. (2023). Besides the climatological and economical consequences, volcanic gases and particles regionally lead to strong pollution events, too, so that during volcanic eruptions villages in the proximity even have to be evacuated. Often, the visibility is reduced and

extremely degraded air quality is caused (Pattantyus et al., 2018). As a dominant component of PM2.5, sulfate aerosol has a negative impact on human health as it infiltrates deeply into the lung and can cause asthma, sinusitis or further respiratory disease (Businger et al., 2015).

Lidar observations have expanded our knowledge on volcanic aerosol in the troposphere. In the case of the eruption of Eyjafjallajökull in Iceland in 2010 (Ansmann et al., 2010, 2011b; Groß et al., 2012; Pappalardo et al., 2013) and Etna in Italy in 2002 (Pappalardo et al., 2004) and 2019 (Kampouri et al., 2021), pure volcanic ash was observed. Lidar ratios in the range of 30–60 sr and a particle linear depolarization ratio of 35–37 % were measured at 355 and 532 nm. In general, it is challenging to distinguish volcanic ash from other depolarizing aerosol types, especially from desert dust, because of the very similar lidar ratios of both types. The main quantity for the distinction between volcanic ash and desert dust is the particle linear depolarization ratio, which is in the range of 30±5 % (at 355, 532, 710 and 1064 nm) for pure dust (Ansmann et al., 2010) and, thus, smaller than the aforementioned values for ash. Sulfate aerosol instead can be distinguished more easily from volcanic ash due to a much lower particle linear depolarization ratio, which is close to zero, and the different size ranges of the aerosol particles. While volcanic ash is in the coarse mode (diameter $> 2\,\mu m$), sulfate aerosol is in the fine mode (diameter $\leq 2\,\mu m$) (John et al., 2011). A separation of volcanic sulfate and ash based on the particle linear depolarization ratio was successfully introduced by Ansmann et al. (2011b). Sulfate particles produce a larger lidar ratio (55 up to 80 sr) and a particle linear depolarization ratio close to zero (4–5 %) as multiwavelength-Raman lidar observations at 355 and 532 nm during the eruptions of Eyjafjallajökull and Etna have shown (Pappalardo et al., 2004; Mona et al., 2012; Navas-Guzmán et al., 2013). In the case of Eyjafjallajökull, Navas-Guzmán et al. (2013) observed two distinct aerosol layers over Granada, Spain, consisting of 82 % of sulfate aerosol. Sulfate aerosol from Eyjafjallajökull mixed with continental aerosol was furthermore observed in the planetary boundary layer (PBL) over Potenza, Italy, (Mona et al., 2012). One of the first multiwavelength-Raman lidar measurements of tropospheric volcanic aerosol, and especially sulfate particles (mixed with a low amount of soot), was also performed at Potenza by Pappalardo et al. (2004), capturing the eruption of Etna in 2002.

One of the most recent volcanic eruptions, which was highly present in the European media, took place at the Cumbre Vieja volcanic ridge (28.62°N, 17.88°W, 1949 m a.s.l.) at La Palma, Canary Islands. The event is described in detail by Carracedo et al. (2022). Further studies concerning its impact on air quality were performed by Filonchyk et al. (2022) and Milford et al. (2023). A ceilometer-based study of the mass concentration of volcanic ash at La Palma and its distribution to the south of France was performed by Bedoya-Velásquez et al. (2022). Volcanic activity started on 19 September 2021. The last eruption was recorded on 13 December 2021. The eruptive column usually reached to an altitude of 3500 m a.s.l. and peaked at 8500 m a.s.l. on 13 December. During the whole time of volcanic activity, fine lapilli (diameter 2–64 mm) were constantly produced. In addition, ash (<2 mm) and more than 10 kt $SO_2$ per day were emitted (Filonchyk et al., 2022) so that at different measurement sites at La Palma the European air quality hourly threshold of $350\,\mu g\,m^{-3}$ was exceeded on multiple days (Milford et al., 2023). The $SO_2$ emissions were largest at the beginning of the period with a maximum of 125 kt on 23 September 2021 (Milford et al., 2023). According to $SO_2$-dispersion forecasts (Carracedo et al., 2022), the emission products were transported over long distances reaching Central Europe and the Caribbean. A mixture of fine and coarse mode aerosol originating from La Palma was detected at Toulouse, France, on 24–25 September 2021 (Bedoya-Velásquez et al., 2022).

Volcanic aerosol of this eruption was also transported towards Mindelo on the Cabo Verdean Islands, which are located 1500 km southwest of the Canary Islands. Since June 2021, the multiwavelength-Raman-polarization lidar Polly$^{XT}$ (Engelmann et al., 2016; Baars et al., 2016) has been operated there and was able to capture the volcanic aerosol plume. On 24 September 2021, the volcanic particles caused a high aerosol optical depth (AOD) of around 1.0, as measured with the co-located sun photometer, and a strong pollution in the PBL with extinction coefficient values more than twice as much as the typical background conditions leading to a highly reduced visibility. We will show that the volcanic aerosol reached the measurement site at a low altitude and, thus, had a significant relevance with regard to air quality and human health. In this paper, we present a case study of lidar observations conducted on 24 September 2021 (period of volcanic activity) at Mindelo, Cabo Verde contrasted to a reference measurement from 16 September 2021, before the start of the eruptions. In the following section, the methodology is described, including information about the instruments and models, the measurement site and the method of data processing. In Sect. 3, the results for the case study are presented and discussed in Sect. 4, before a conclusion is drawn in Sect. 5.

## 2 Methodology

### 2.1 Measurement site and instrumentation

In the frame of the ground-based part, called ASKOS (it is a Greek word originating from the Greek mythology and only denotes the name of the campaign), of the Joint Aeolus-Tropical Atlantic Campaign (JATAC) (Amiridis et al., 2022; Fehr et al., 2023; Marinou et al., 2023), a temporary Aerosol, Clouds and Trace Gases Research Infrastructure (ACTRIS) remote sensing station was set up at the Ocean Science Center Mindelo (OSCM) at Mindelo, Cabo Verde, (16.878°N, 24.995°W) in June 2021. The OSCM is located on the west coast of the island of São Vicente, with low anthropogenic influence. The island itself is located 1500 km southwest of the Canary Islands and La Palma and in the trade wind zone with usual advection of air masses from north-easterly direction. Typically, cumulus convection occurs at Mindelo during nighttime.

Amongst others, this station is equipped with a Polly$^{XT}$ multiwavelength-Raman-polarization lidar. The lidar deployed at Mindelo has a few improvements, compared to previous instruments (Althausen et al., 2009; Engelmann et al., 2016). For instance, it uses a diode pumped Nd:YAG laser, which has a higher repetition rate (100 Hz) than the typical flashlamp pumped Nd:YAG laser (20–30 Hz) of Polly$^{XT}$. This feature offers the possibility to retrieve profiles of the optical properties with a lower temporal averaging down to 10 min. It is an important capability with regard to this study, since at Mindelo small clouds often occur at night and cloud-free periods are quite short. Furthermore, the receiver consists of 15 channels and enables measurements of the elastic backscatter coefficient at 355, 532 and 1064 nm, the inelastic backscatter at 387, 607 and 1058 nm, the cross-polar signal at 355, 532 and 1064 nm and the inelastic signal from water vapour at 407 nm. Additionally to the far-field (ff) measurements, near-field (nf) measurements are available for the 355 and 532 nm elastic channels and the 387 and 607 nm Raman channels. The instrument has also a dual-field-of-view depolarization channel (Jimenez et al., 2020a), which is a powerful technique, allowing the determination of microphysical liquid-water properties (Jimenez et al., 2020b). Thus, in combination with the lidar-derived aerosol optical properties, it can be used to study aerosol-cloud interactions, which is,

however, not the scope of the study we are presenting here. With the described setup, several aerosol optical properties can be determined. These are the particle backscatter coefficient, the particle extinction coefficient, the lidar ratio (ratio of particle extinction-to-backscatter coefficient) and the particle linear depolarization ratio, all at 355, 532 and 1064 nm, as well as the backscatter-related Ångström exponent between the different wavelengths and the extinction-related Ångström exponent. The availability of the extinction coefficient, the lidar ratio and the particle linear depolarization ratio at 1064 nm as well as the backscatter coefficient at this wavelength determined via the rotational Raman (RR) channel is a new feature of this device. The calculation of the extinction coefficient at 1064 nm via the rotational Raman method follows the methodology described in Haarig et al. (2016). The spectral cross-talk calibration using a liquid cloud with a constant cloud base height was introduced in Haarig et al. (2022). Here, a liquid cloud base on 4 October 2021 was used for calibration, which led to a spectral cross-talk correction factor of 6.7e-4±0.3e-4. There was no change in neutral density filters between 24 September and 4 October 2021 and therefore the spectral cross-talk correction factor remained stable. The calibration of the depolarization ratio at 1064 nm and the estimation of its uncertainties followed the same approach as the calibration at 355 and 532 nm (Engelmann et al., 2016). The Δ90° calibration (Freudenthaler et al., 2009) with a linear polarizer after the pinhole was applied. Like every Polly$^{XT}$ lidar system, the Polly$^{XT}$ lidar that operates at Mindelo is part of Polly$^{NET}$ (Baars et al., 2016) and vertical profiles of the lidar optical properties are automatically derived by the Polly$^{NET}$ processing chain (Yin and Baars, 2021). However, for this study, the profiles were analyzed manually. Due to the frequent occurrence of boundary layer clouds, a more tailored data analysis was needed. Furthermore, to reduce noise, the profiles were smoothed vertically by using a moving average filter. The result is again a continuous profile with the distance of 7.5 m between the single data points but starting at an altitude, which is half of the smoothing length ($s$). Thus, each data point contains information of the height range from $0.5 \cdot s$ below to $0.5 \cdot s$ above this point.

Concerning the uncertainties of the lidar-derived aerosol optical properties, it is worth to mention that systematic errors (e.g., polarization effects in the receiver unit, dead time effects, overlap effects) are generally corrected as the lidar system is calibrated according to ACTRIS/EARLINET standards (e.g., ACTRIS guidelines, 2024). To account for remaining unknown systematic errors (e.g. reference height and value) and the statistical uncertainties, a relative error of 15 % is considered for the particle backscatter coefficient determined with the Raman method (Althausen et al., 2009; Baars et al., 2012; Engelmann et al., 2016; Baars et al., 2016). For the particle extinction coefficient, the statistical error is calculated from the error of the linear fit of the derivative without considering systematic uncertainties. This linear fit considers as much data points as the smoothing length ($s$) and is applied every 7.5 m. For the particle depolarization ratio, we consider a remaining constant absolute error of 0.02 at 355 nm and of 0.01 at 532 and 1064 nm retrieved after intense calibration approaches (ACTRIS guidelines, 2024). The uncertainties of the lidar ratio and the Ångström exponents were then calculated using the Gaussian error propagation.

In addition, a CIMEL Sun Sky Lunar photometer of type CE318-T was used for this study, which is operating in the Aerosol Robotic Network (AERONET; Holben et al., 1998). It measures solar irradiances at 8 different wavelengths (340, 380, 440, 500, 675, 870, 1020 and 1640 nm) from which the AOD (at the same wavelengths), the columnar Ångström exponent (for 6 wavelength pairs), the volume size distribution, the refractive index, the single scattering albedo, the absorption AOD, the

extinction AOD, the asymmetry factor, and the phase function are derived. A new capability of the latest type CE318-T is that it measures during night as well using the moonlight to determine nighttime AODs.

## 2.2 Air mass source attribution

To describe the origin of the observed air masses, Hybrid Single-Particle Lagrangian Integrated Trajectories (Stein et al., 2015; Rolph et al., 2017; HYSPLIT, 2024) were used. Ensemble trajectories with 27 members were calculated for 5 days back in time, i.e., towards the day when volcanic activity at La Palma started. The meteorological input data was taken from the Global Data and Assimilation Service (GDAS1, 2024). Furthermore, simulations with the air mass source attribution tool TRACE (Radenz et al., 2021; Radenz, 2021) were performed, which is a combination of the FLEXible PARTicle dispersion model FLEXPART (Pisso et al., 2019) and a simplified version of the MODIS land cover classification (Broxton et al., 2014) or custom defined geographical areas. For FLEXPART, the meteorological input data was taken from the Global Forecast System (GFS; National Centers for Environmental Prediction, National Weather Service, NOAA, U.S. Department of Commerce (2000)). In this case, 5-day backward simulations were calculated for 500 air parcels, which arrive at Mindelo at different altitudes from 0 to 10 km in steps of 500 m with a temporal resolution of 3 h. Evaluating both backward simulation models allows us to ensure more certainty with respect to the origin of the air masses.

In addition, the horizontal distribution of the volcanic plume was monitored. Therefore, the transport of $SO_2$ and its advection towards Mindelo was tracked. For this purpose, the TROPOspheric Monitoring Instrument (Veefkind et al., 2012; TROPOMI, 2024) on board the polar orbiting Sentinel-5 Percursor satellite was used, which offers daily global measurements of the amount of $SO_2$ molecules in a column per surface area. Its horizontal resolution is $3.5 \times 5.5 \, km^2$.

## 3 Results

A time series of the AERONET AOD at different wavelengths and the columnar Ångström exponent between 440 and 870 nm is shown in Fig. 1. Before the start of the eruption, the hourly mean AOD was around 0.4 at the shown wavelengths. Hourly mean Ångström exponent values of 0.2 were usually observed until 22 September. During the time of volcanic activity, a change in the behaviour of the Ångström exponent and the AOD could be seen since 22 September and, thus, 3 days after the eruption started. A strong increase of the Ångström exponent to values higher than 0.8 was measured. In that time, high AOD values close to 1.0 at wavelengths $\leq 500$ nm were recorded, e.g., on 24 and 29 September 2021. On 24 September, the daily mean AOD was 1.1 at 340 nm and 0.9 at 500 nm. The vertically-resolved lidar optical properties are presented in a case study for 24 September (Sect. 3.2) contrasted to background conditions before the volcanic eruption (16 September, Sect. 3.1), representing a clean PBL (marine influenced).

## 3.1 Reference case (16 September 2021)

To contrast the differences between the volcanic influenced aerosol conditions over Mindelo and the typical situation before the start of the eruption at La Palma, the 16[th] September 2021 was selected as reference observation. The corresponding height-

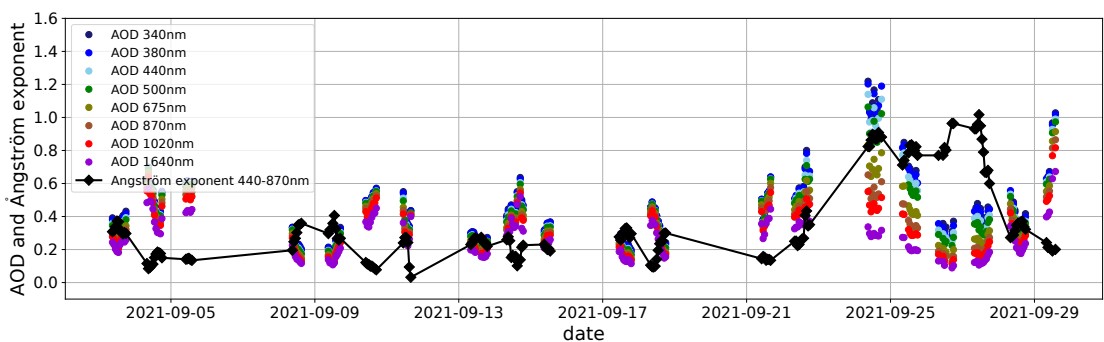

**Figure 1.** Time series of Level 2.0 hourly averages of the aerosol optical depth (AOD) at different wavelengths and the columnar Ångström exponent between 440 and 870 nm measured with an AERONET sun photometer at Mindelo in September 2021. Data points for the 16 September were cut out, since on that day, a cirrus was present, which was not correctly screened out by the AERONET retrieval. The original data, including the cirrus, is shown in the Appendix (Fig. A1).

resolved temporal development of the calibrated attenuated backscatter coefficient at 1064 nm and the volume depolarization ratio at 532 nm is shown in Fig. 2 (left side). The figure is provided to illustrate the vertical structure of the aerosol layers for the days of interest. As it is not corrected for the atmospheric attenuation, it does not allow a quantitative comparison of the backscatter intensity at a given altitude. For this purpose, vertical profiles of the backscatter coefficient are presented later on. The vertical structure on 16 September showed two different aerosol layers. The PBL reached up to 0.8 km height. In that layer, no depolarization occurred (Fig. 2b). Above, a lofted layer, which was strongly depolarizing, was located between 1.2 and 6 km height. Small clouds were frequently present in the PBL as indicated by a very strong backscatter signal and complete attenuation (no signal) some few hundreds of meters above the cloud base. Such a vertical structure has been typically observed over Mindelo from June to October 2021 and is in agreement with previous studies on this archipelago (Ansmann et al., 2011a; Groß et al., 2011; Rittmeister et al., 2017). Additionally, on the reference day, a cirrus occurred, which was not correctly screened out by the AERONET algorithm (cf. Fig. A1 and Fig. A2). Thus, there are no usable sun photometer data for 16 September.

Vertical profiles of the lidar optical properties (Fig. 3) were derived with the Raman method (Ansmann et al., 1992) for a 48-min interval in the evening (22:24–23:12 UTC, red rectangle in Fig. 2, left side), since this was the longest cloud-free period during nighttime. The corresponding mean values are summarized in Table 1. The uncertainties given in the text are always the standard deviation (parameter variability within the layer) or the layer mean error as described in the caption of Table 1. In the lofted layer, between 1.3 and 5.3 km height, the mean lidar ratio of $58.4 \pm 8.8$ and $47.3 \pm 7.2$ sr (at 355 and 532 nm) and the particle linear depolarization ratio of $24.5 \pm 2.0$, $28.1 \pm 1.0$ and $24.1 \pm 1.0$ % (at 355, 532 and 1064 nm) are in a typical range for desert dust (Haarig et al., 2017; Floutsi et al., 2023). In the PBL (up to 0.6 km height), relatively clean marine conditions (Bohlmann et al., 2018) were observed, which were characterized by a mean lidar ratio of $17.3 \pm 2.8$ and $23.8 \pm 4.2$ sr (at 355 and 532 nm) and a mean particle linear depolarization ratio of $\leq 1.1$ % (at 355, 532 and 1064 nm). The mean particle extinction

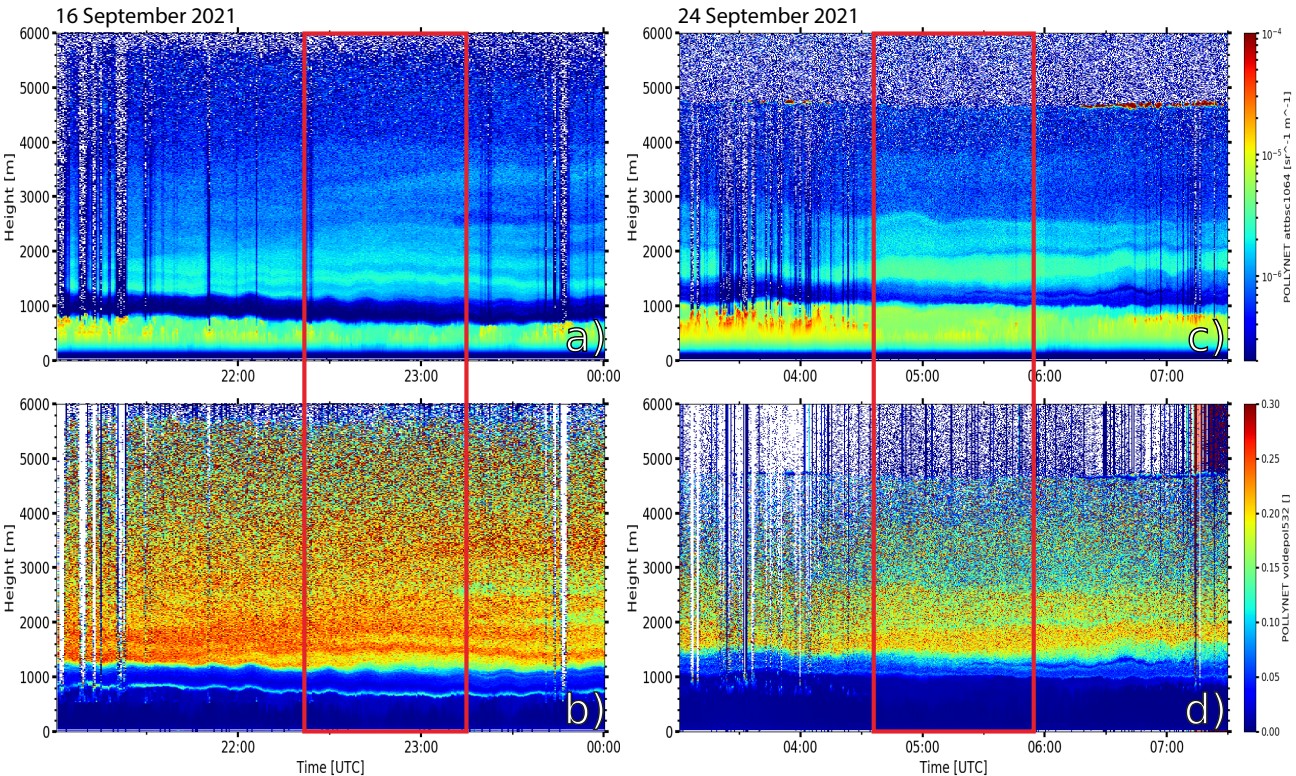

**Figure 2.** Temporal evolution of the height-resolved calibrated attenuated backscatter coefficient at 1064 nm (top) and the volume depolarization ratio at 532 nm (bottom) measured by Polly$^{XT}$ at Mindelo, Cabo Verde, during 16 September 2021, 21:00–24:00 UTC (left) and 24 September 2021, 03:00–07:30 UTC (right).

coefficient was about $114\pm20$ and $130\pm24$ Mm$^{-1}$ (at 355 and 532 nm). Unfortunately, for that day the rotational Raman profiles at 1064 nm were not available since the analyzed time period of 48 min is too short to obtain reasonable results. However, the
210 measurement from 16 September represents the typical values, which we usually observed over Mindelo during that time of the year, as the lidar studies in the framework of ASKOS and L2A+ confirm (L2A+, 2024; EVDC, 2024). This statement is valid especially for the PBL.

### 3.2 Volcanic influence (24 September 2021)

For 24 September 2021, the height-resolved temporal development of the attenuated backscatter coefficient at 1064 nm is
215 shown in Fig. 2c and the volume depolarization ratio at 532 nm in Fig. 2d. Again, two distinct aerosol layers are visible – a very low depolarizing PBL (Fig. 2d) up to about 1 km height and a strongly depolarizing lofted layer from 1.4 to 5 km height. As on 16 September, small clouds occurred frequently in the PBL. Before first daylight appeared at 07:30 UTC, a longer cloud

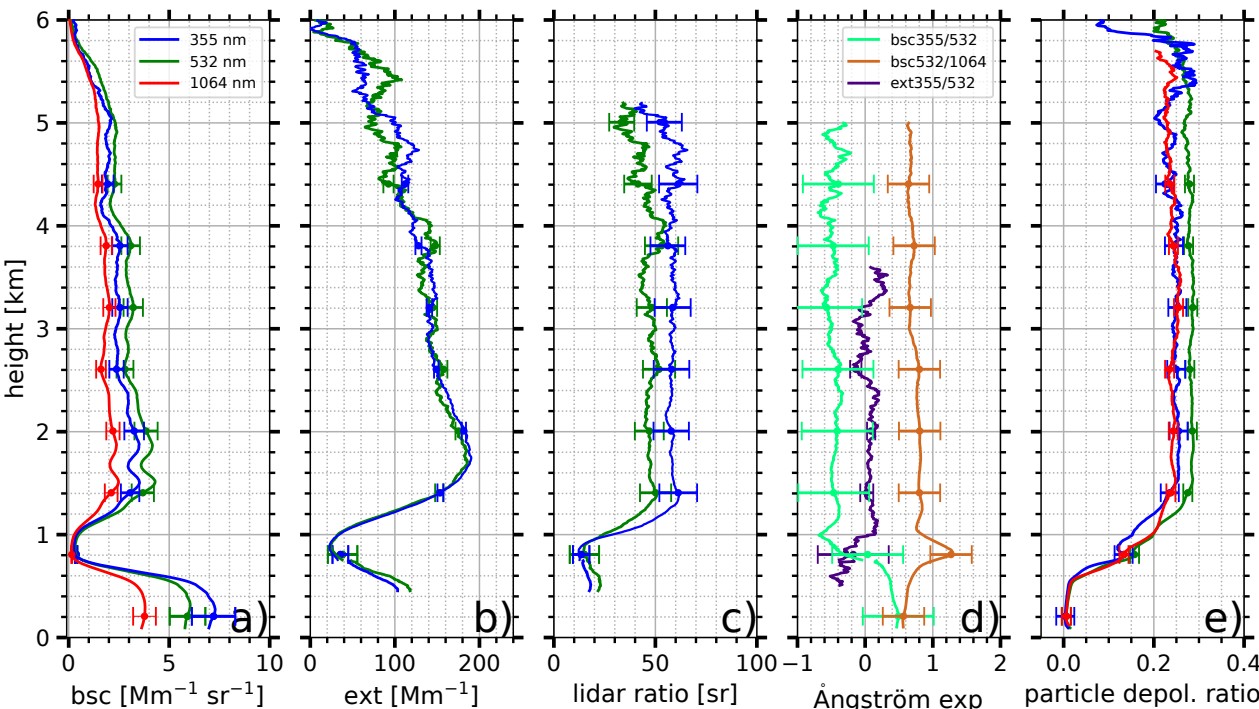

**Figure 3.** Measured with Polly[XT] at Mindelo, Cabo Verde, on 16 September 2021, between 22:24 and 23:12 UTC: vertical profiles of (a) the particle backscatter coefficient, (b) the particle extinction coefficient, (c) the lidar ratio, (d) the Ångström exponent and (e) the particle linear depolarization ratio. Vertical smoothing: 187.5 m for (a), (e) and the backscatter-related Ångström exponent and 742.5 m for (b), (c) and the extinction-related Ångström exponent. Near- and far-field measurements are merged at 750 m. The error bars show the uncertainties described in Sect. 2.1.

free period evolved. Thus, optical properties were retrieved with the Raman method for an 1:19 h interval (04:38–05:57 UTC, indicated by a red rectangle in Fig. 2, right side).

The corresponding vertical profiles are shown in Fig. 4. On that day, all lidar-derived optical quantities are available at all three wavelengths. For the lofted layer, mean values, as depicted in Table 1, were retrieved based on the far-field measurements. The particle extinction coefficient was in the range of 114–168 Mm$^{-1}$ at 355, 532 and 1064 nm. Measurements of the lidar ratio led to layer mean values of $64.8 \pm 10.2$, $50.9 \pm 8.3$ and $61.8 \pm 8.6$ sr (355, 532, and 1064 nm, respectively). These values are slightly larger than the ones measured on 16 September. From 532 nm to 1064 nm, the lidar ratio increased by 21 %, which is in
line with dust observations at Leipzig, Germany (increase by 24–38 %; Haarig et al., 2022). Because of similar source regions of the dust, namely the Western Sahara, both observations are comparable. The higher lidar ratio at 355 nm compared to 532 nm suggests specific source regions in the Sahara, as observed through lidar observations in Senegal (Veselovskii et al., 2020). The

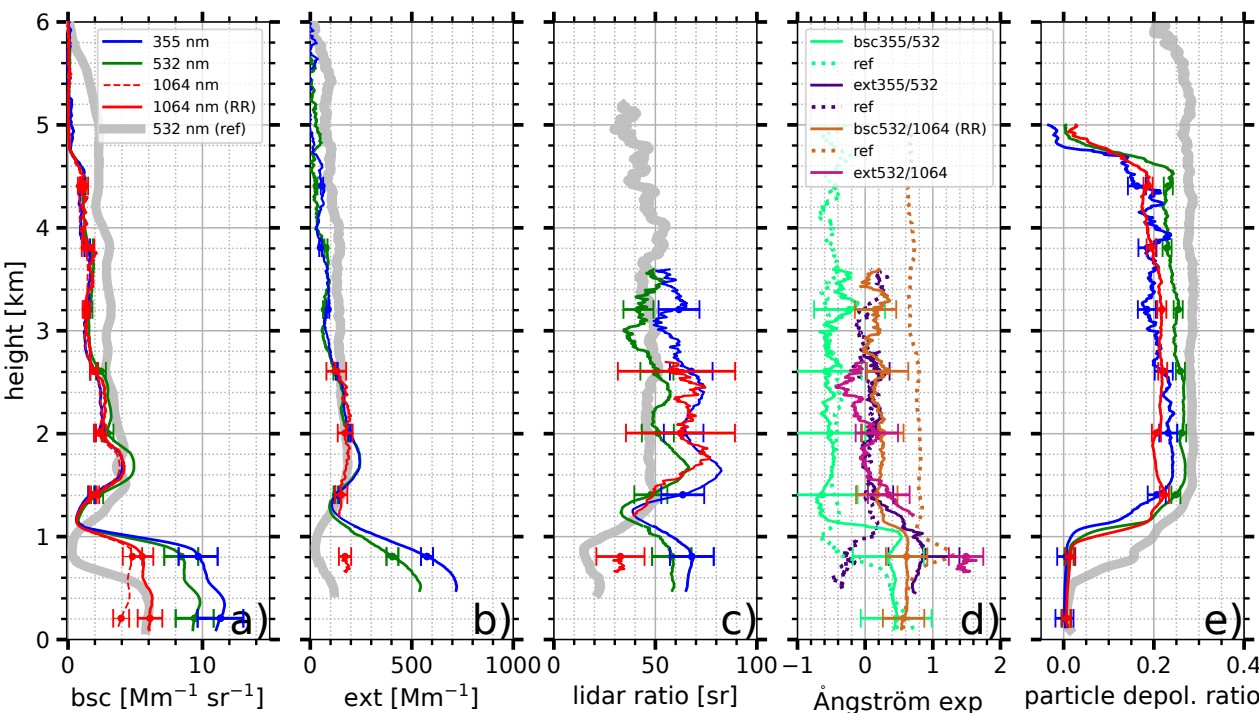

**Figure 4.** Same as in Fig. 3 but for the 24 September 2021, between 04:38 and 05:57 UTC (note the changed x-scale for (a) and (b)), including the particle extinction coefficient and lidar ratio at 1064 nm and the extinction-related Ångström exponent between 532 and 1064 nm (vertical smoothing: 397.5 m below 1.2 km and 1492.5 m above 1.2 km). Near- and far-field measurements are merged at 1100 m. Reference profiles at 532 nm from 16 September 2021 are shown as thick grey line and in (d) as dotted lines and labelled as "ref".

measured particle linear depolarization ratio of 20.6–25.0 % for the three different wavelengths indicates the presence of non-spherical particles, i.e., desert dust, but is somewhat smaller than what was typically observed for pure dust (Freudenthaler et al., 2009; Floutsi et al., 2023), indicating the presence of some spherical non-dust particles. Considering the wavelength dependence of the particle linear depolarization ratio, a relative decrease of 18 % from 532 towards 1064 nm was observed. Similar findings were made at Leipzig, Germany, and Morocco during SAMUM (relative decrease by 13–31 %; Freudenthaler et al., 2009; Haarig et al., 2022). The backscatter-related Ångström exponent in the lofted layer is on average around $0.4 \pm 0.31$ for the wavelength pair 532/1064 nm, indicating large particles (i.e., desert dust). Considering the higher lidar ratio (especially at 355 nm) and the lower particle linear depolarization ratio on 24 September compared to the typical values of pure desert dust, we conclude that the dust on 24 September was slightly polluted.

In contrast to the almost typical aerosol conditions in the lofted layer, an unusual strong pollution was observed in the PBL. The findings are highlighted in Fig. 5 showing zoomed profiles. In addition, vertical smoothing was reduced, which improves

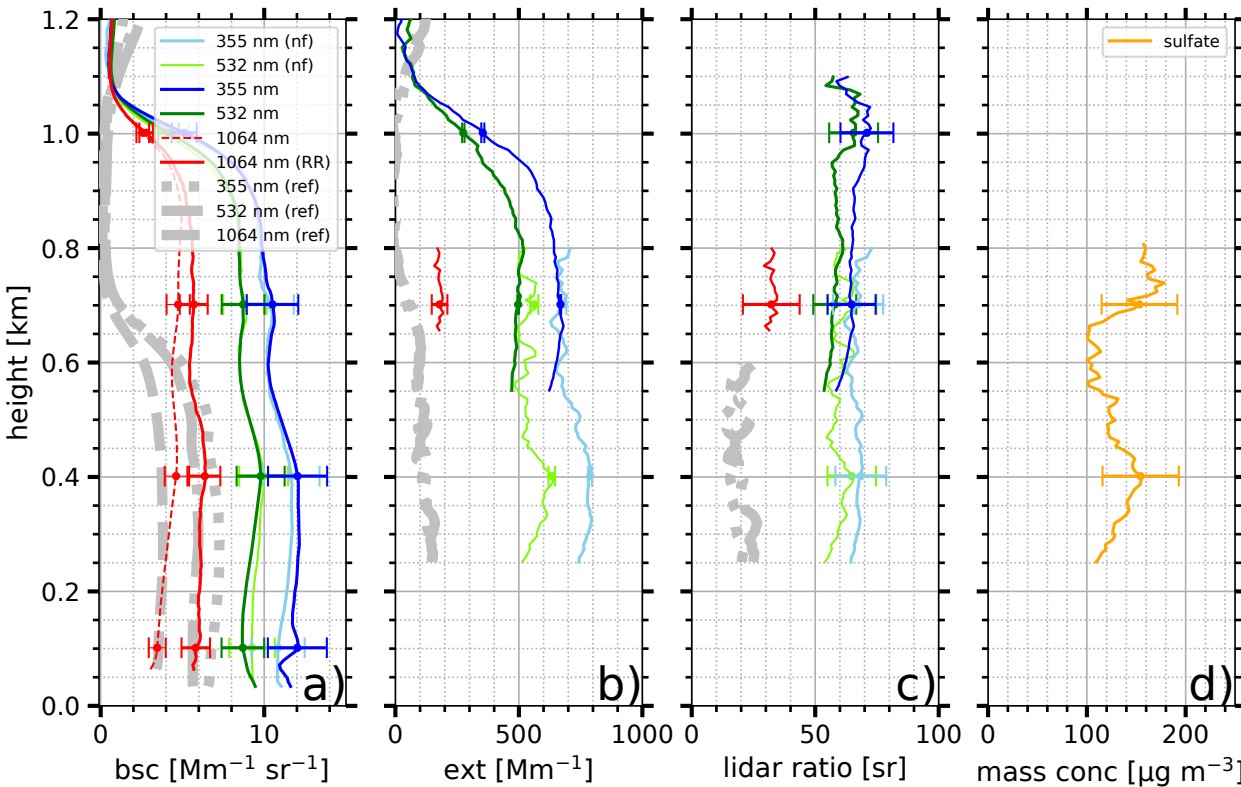

**Figure 5.** (a)–(c): same profiles as in Fig. 4 (a)–(c), but only up to a height of 1.2 km. Vertical smoothing: (a) 67.5 m, (b) and (c) 187.5 m at 355 and 532 nm and 742.5 m at 1064 nm. Near- and far-field measurements are shown separately. (d): sulfate mass concentration with height constant relative error of 25 %.

the resolution, reduces the overlap effect especially for the near-field profiles and, thus, allows to include information from lower altitudes above the lidar. All mean values for the PBL are listed in Table 1 as well. On this day, extremely high values of the particle extinction coefficient were observed with layer mean values of $721 \pm 51$, $549 \pm 38$ and $178 \pm 13$ Mm$^{-1}$ (at 355, 532 and 1064 nm) in the PBL. The maximum values were even higher with $794 \pm 7$, $640 \pm 13$ and $198 \pm 26$ Mm$^{-1}$ (at 355, 532 and 1064 nm, mentioned here with statistical errors). These values are 3–4 times higher than what was observed under clean marine conditions as shown for 16 September and indicated as grey lines in Fig. 5. Additionally, the particle extinction coefficient was strongly decreasing with increasing wavelength. A similar behavior was observed for the lidar ratio. Mean values of $66.9 \pm 10.1$, $60.2 \pm 9.2$ and $30.8 \pm 8.7$ sr (at 355, 532 and 1064 nm) were found, showing a decrease by 48 % from 532 towards 1064 nm. The mean values of the lidar ratio are notably high compared to the clean marine conditions and are typical for pollution or even smoke (Floutsi et al., 2023). However, the decrease of the lidar ratio at 1064 nm compared to the value at 532 nm points

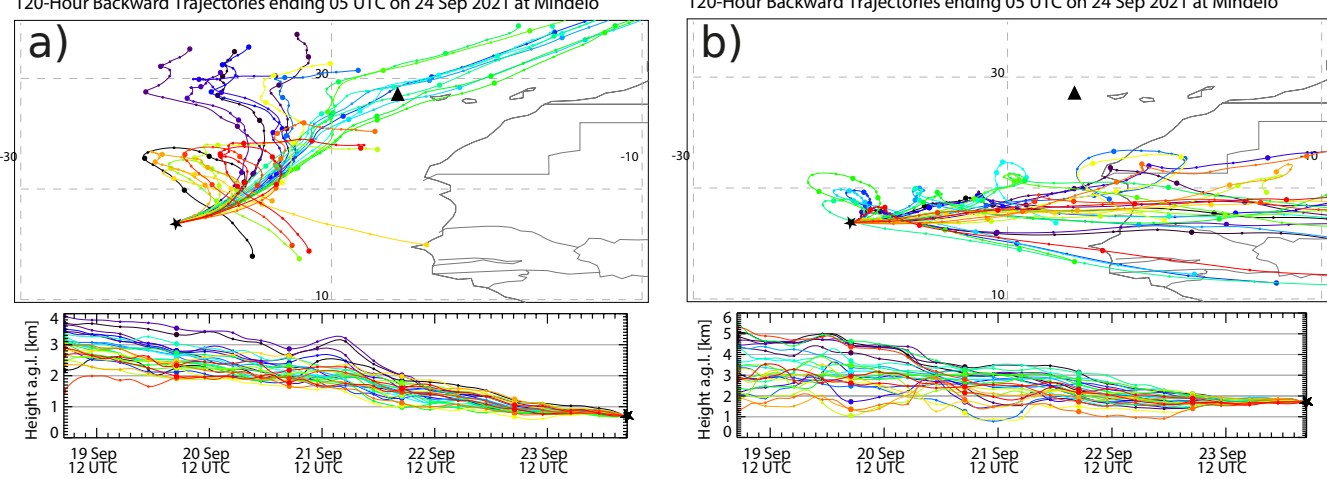

**Figure 6.** HYSPLIT ensemble trajectories for 120 hours back in time are shown. Backward trajectories of air masses arriving at Mindelo (black star) on 24 September 2021, 5 UTC at (a) 0.7 km and (b) 1.7 km were computed. In the lower altitude, they mainly originate from La Palma (black triangle), whereas in the higher altitude, they were advected from the Sahara.

rather to pollution than to smoke. In the case of wildfire smoke an increase of the lidar ratio at 1064 nm was observed (Haarig
et al., 2018). The high lidar ratio values point out the presence of particles, which are strongly attenuating the incoming solar
radiation by scattering and absorption (Wandinger et al., 2023). The large extinction close to the ground indicates a strong
pollution and explains the unusual high daily mean AOD of 1.1, 0.9 and 0.5, which was measured with the sun photometer at
340, 500 and 1020 nm on this day. The lidar-derived total AOD between 04:38 and 05:57 UTC was $0.96 \pm 0.28$, $0.79 \pm 0.26$
and $0.57 \pm 0.17$ at 355, 532 and 1064 nm, respectively, and, thus, in agreement with the values of the sun photometer measured
during daytime. The AOD for the boundary layer only (including the sulfate and marine contribution), as derived from the lidar
measurements, was $0.58 \pm 0.03$, $0.43 \pm 0.02$ and $0.18 \pm 0.01$ at 355, 532 and 1064 nm and, thus, covered 54–60 % of the total
lidar-derived AOD in case of 355 and 532 nm and 32 % in case of 1064 nm. Furthermore, visibility was strongly reduced on
that day. Based on the maximum particle extinction coefficient at 532 nm and using the Koschmieder equation (Koschmieder,
1924), we calculated the visibility to be around 6 km. The presence of relatively small particles is indicated by the moderate
wavelength dependence represented by the backscatter-related Ångström exponent between 532 and 1064 nm (RR), which
was $0.54 \pm 0.31$ and the mean extinction-related Ångström exponent of only $0.68 \pm 0.07$. The low values of the particle linear
depolarization ratio $\leq 0.9$ % indicate that the observed particles were spherical.

## 4   Discussion

To summarize, both days – the 16[th] September 2021 (before the start of the eruption at La Palma) and the 24[th] September
2021 (during the volcanic eruption episode) – had a similar aerosol layering structure with a PBL $\leq 1$ km and a lofted layer

**Figure 7.** Satellite observations of TROPOMI on Sentinel-5P, which show the column integrated of $SO_2$ mass for the Cabo Verdean region on 23 September 2021 (TROPOMI, 2024).

of Saharan dust up to 6 km, typical for this time of the year at Cabo Verde. Both measurements were taken under similar meteorological conditions and, thus, well suitable to assess the influence of the volcanic activity.

Although for the lofted layer, the lidar-derived aerosol optical properties slightly vary concerning the extent and intensity between 16 and 24 September 2021 with a lower layer top height, a lower particle linear depolarization ratio and a higher lidar ratio on 24 September, we can conclude that on both days the predominant aerosol type in the lofted layer was Saharan dust. Obviously, no volcanic ash was included in this layer on 24 September, because in that case we would have observed much higher values of the particle linear depolarization ratio (Groß et al., 2012). The higher lidar ratio observed on 24 September, however, indicates the presence of stronger absorbing particles slightly contaminating the Saharan Air Layer (SAL) on that day. Considering the particle linear depolarization ratio and the lidar ratio together, a contamination with continental pollution or smoke is feasible. To corroborate the origin of the lofted aerosol layer on this day, 120 h HYSPLIT ensemble backward trajectories are shown in Fig. 6. Simulations of air mass arrival at Mindelo on 24 September, 5 UTC, at 0.7 km (Fig. 6a) and at 1.7 km (Fig. 6b) have been calculated. The trajectories for the higher altitude (Fig. 6b) show that the lofted layer was

influenced by an easterly flow so that air masses were advected directly from the Saharan desert, which makes the occurrence and predominance of Saharan dust evident. However, partial mixing with sulfate during the transport over the Atlantic Ocean cannot be ruled out as well as smoke and pollution contamination over the African continent. Fire spot analysis with the Fire Information for Resource Management System (FIRMS, 2024, Fig. B1) revealed only little fire activity along the transport path within the 120 h. Fires were detected at the eastern border of Algeria and close to the Mediterranean. Thus, smoke contamination may have led to the slight contamination of the SAL. However, we consider the presence of volcanic ash based on this analysis and the eruption mechanisms at Cumbre Vieja to be unlikely.

In contrast, the aerosol conditions in the PBL strongly differed between both analyzed measurement periods. While on 16 September, a clean marine PBL was present, a strong pollution was observed on 24 September with layer mean values of the particle extinction coefficient up to almost $800\,\mathrm{Mm^{-1}}$, compared to $\leq 130\,\mathrm{Mm^{-1}}$ during the clean marine conditions. The lidar ratio on that day was strongly enhanced with values around $60\,\mathrm{sr}$ compared to values for pure marine conditions of around $20\,\mathrm{sr}$. The aerosol load in the PBL was furthermore responsible for 54–60 % of the total AOD at 355 and 532 nm and for 32 % at 1064 nm in the analyzed time period. In contrast, for the 16 September, the contribution of the AOD in the PBL to the total AOD was only 12 % (at 355 and 532 nm), i.e., more than 80 % of the total AOD was caused by the SAL. The observed pollution is associated to air masses coming from La Palma containing volcanic aerosol, which is supported by the HYSPLIT backward trajectories depicted in Fig. 6a. They illustrate a distinct advection of air masses from Canary Islands and, thus, from the volcano on La Palma. Additionally, TRACE backward simulations (Fig. C1) confirm these findings as they show air masses accumulating over the Atlantic Ocean on 18 September before they pass from northwest over La Palma on 21 September and move further to Mindelo on 23 September. TRACE backward simulation were also performed for air parcels arriving at Mindelo at 500 m altitude on 16 September 2021, 0 UTC, shown in Fig. C2. The pathway of the air parcels was similar to the one from 24 September. While on 16 September, they also partly crossed the African continent, they passed only over the Atlantic Ocean and the Canary Islands on 24 September. This finding emphasizes even more the impact of the volcanic eruption on the pollution observed in the PBL at Mindelo. As the corresponding particle linear depolarization ratio is low, the presence of ash particles can be excluded.

Instead, volcanic sulfate seemed to be the dominating aerosol type in the PBL. It becomes more evident if the large amount of sulfur dioxide released by the volcano is taken into account. The $SO_2$ emissions were greatest at the beginning of the active period, reaching a maximum of 125 kt on 23 September 2021 (Milford et al., 2023). $SO_2$ was advected towards Mindelo as can be seen in the satellite measurements of TROPOMI onboard Sentinel-5P (Fig. 7), showing the amount of $SO_2$ around the Cabo Verdean region during afternoon of 23 September 2021. The presence of $SO_2$ offered the possibility for secondary aerosol formation to sulfate particles, which is assumed to be the source of the observed particles. $SO_2$ quickly oxidates to sulfate aerosol with a high efficiency at warm temperatures and high relative humidity (Eatough et al., 1994; Yang et al., 2018). Favorable conditions seemed to be prevalent since the air masses were transported only over the Atlantic Ocean in a tropic region. According to Pattantyus et al. (2018), conversion rates are large ($3–50\,\%\mathrm{s^{-1}}$) especially in cloudy air, which is given due to the frequently occurring small clouds in the PBL as observed over the Cabo Verdean region.

In addition, not only SO$_2$ could have been advected from La Palma but also sulfate particles themselves. Filonchyk et al. (2022) identified, based on the single scattering albedo and the dissection of the AOD into a coarse and fine mode component, that on 24 September 2021 coarse-mode particles were almost absent at La Palma. Instead the presence of non-absorbing but scattering fine-mode particles attributed to sulfate aerosol was shown, which could have been formed locally or were emitted directly by the volcano. As these observations are valid for a time period in which our case study was included, it strengthens our hypothesis that we measured volcanic sulfate at Mindelo originating from Cumbre Vieja.

The presence of sulfate aerosol from the volcanic eruption at La Palma also becomes evident since the measured aerosol optical properties are in agreement with previous lidar observations of volcanic sulfate (e.g., Pappalardo et al., 2004; Mona et al., 2012; Navas-Guzmán et al., 2013). Furthermore, in this study, it was the first time ever that tropospheric volcanic sulfate was measured with a lidar at 1064 nm. During the eruption of Eyjafjallajökull in 2010, Navas-Guzmán et al. (2013) observed lofted aerosol layers between 1.5 and 3.5 km consisting to 82 % of fine-mode aerosol particles, i.e., sulfate particles, over Granada, Spain. The corresponding values of the lidar ratio were 55 and 75 sr (355, 532 nm). Mona et al. (2012) recorded values of the lidar ratio up to 80 sr for volcanic sulfate from Eyjafjallajökull mixed with continental aerosol in the PBL over Potenza, Italy, while during the eruption of Mt. Etna in 2002 a lidar ratio of $55 \pm 4$ sr (355 nm) was measured by Pappalardo et al. (2004) in a lofted aerosol layer of young sulfate particles mixed with a low amount of soot between 4 and 4.5 km over Potenza. With $66.9 \pm 10.1$ and $60.2 \pm 9.2$ sr (355, 532 nm), the observations over Mindelo on 24 September fit well into the range of values of the lidar ratio observed during the previous eruptions. The observed wavelength dependence with decreasing lidar ratio by 48 % from 532 towards 1064 nm confirms the assumptions in the CALIPSO aerosol characterization which uses a lidar ratio of 50 sr at 532 nm and of 30 sr at 1064 nm for sulfate (Kim et al., 2018; Tackett et al., 2023). The particle linear depolarization ratio measured on 24 September 2021 was with $\leq 0.9$ % even smaller than the values observed for sulfate from Eyjafjallajökull, which was 4–5 % (Navas-Guzmán et al., 2013), indicating more clearly the presence of spherical (sulfate) particles. For the volcanic sulfate from Eyjafjallajökull, the backscatter-related Ångström exponent measured over Granada was $1.1 \pm 0.2$ for the the wavelength pair 355/532 and $2.1 \pm 0.1$ for 532/1064. During the measurement period, the values decreased to $0.7 \pm 0.1$ and $1.7 \pm 0.3$, respectively. Navas-Guzmán et al. (2013) stated that the temporal evolution in the Ångström exponent arose from an increasing particle size in the accumulation mode driven by hygroscopic growth and a potential change in the chemical composition rather than an increasing contribution of coarse-mode particles. In a second layer of volcanic sulfate, values of $1.7 \pm 0.1$ (355/532) and $1.4 \pm 0.2$ (532/1064) were observed by Navas-Guzmán et al. (2013), also decreasing significantly during the measurement period. Compared to these studies, the backscatter-related Ångström exponent measured over Mindelo on 24 September 2021 was smaller having values of only $0.42 \pm 0.52$ and $0.54 \pm 0.31$ (355/532, 532/1064 (RR)). These low values can be explained by hygroscopic growth of the sulfate particles since they were exposed to high humidity during their transport over the Atlantic Ocean before they reached Mindelo. Furthermore, we expect that some marine particles were present in the PBL above Mindelo, which are also spherical at high relative humidity (accounting to the low depolarization ratio). As marine particles are larger than the sulfate particles, they reduced the backscatter-related Ångström exponent in contrast to the aforementioned observations whereat air masses were influenced by the European continent. The extinction-related Ångström

exponent between 355 and 532 nm observed over Mindelo was $0.68 \pm 0.07$ and, thus, closer to the previous observations at Granada whereat values of $0.7 \pm 0.1$ and $0.8 \pm 0.1$ were observed.

As a further step, an attempt to estimate the mass concentration of the observed sulfate aerosol was done according to the method used in Ansmann et al. (2011b). A sulfate conversion factor of $0.2 \cdot 10^{-6}$ was obtained leading to the vertical profile of the mass concentration, which is shown in Fig. 5d, and a layer mean value for the PBL of about $133.1 \pm 20.3\,\mu\text{gm}^{-3}$. Such a high sulfate concentration indicates extremely polluted conditions. For comparison, the aerosol pollution level should not exceed $50\,\mu\text{gm}^{-3}$ in European cities to avoid unhealthy situations. For our estimation, the error of the sulfate mass concentration was assumed to be similar to the one of the dust mass concentration, which is about $20 - 30\,\%$ (Ansmann et al., 2019). Thus, for our estimation, we used a height constant relative error of $25\,\%$.

In this paper, it is the first time ever that we can report the optical properties for the volcanic plume mixed in the marine PBL for all three (aerosol lidar) wavelengths by extending the observational capabilities towards 1064 nm. While the lidar ratios at 355 and 532 nm are in agreement with previous observations, the lidar ratio at 1064 nm of $30.8 \pm 8.7\,\text{sr}$ and the extinction-related Ångström exponent of $1.53 \pm 0.26$ between 532 and 1064 nm have never been reported so far. Thus, it is a milestone for the characterization of volcanic sulfate with remote sensing techniques.

## 5 Summary and conclusions

In the frame of ESA's JATAC campaign to validate the Aeolus satellite, the multiwavelength-Raman-polarization lidar Polly$^{\text{XT}}$ was installed at Mindelo, Cabo Verde, in June 2021 together with further instruments, e.g., an AERONET sun photometer. During the intensive observation period of the campaign in September 2021, a volcanic eruption at the Cumbre Vieja ridge at La Palma, Canary Islands, took place, starting on 19 September 2021. Volcanic activity was recorded until 13 December 2021. Due to the location of Mindelo in the trade wind zone, the preferred wind direction is northeast, i.e., from the Canary Islands. Thus, advected air masses contaminated with volcanic aerosol were observed within the local PBL while the SAL above seemed little affected. The occurrence of volcanic aerosol at Mindelo was indicated by an increase of the columnar Ångström exponent and the AOD as measured by the sun photometer after 22 September 2021. Volcanic aerosol was furthermore observed with the Polly$^{\text{XT}}$ lidar, which is shown in a case study of 24 September 2021. On that day, a pronounced pollution was seen over Cabo Verde, strongly contrasting the conditions observed before the start of the eruption. The intense pollution caused an unusually high AOD of around 1.0 at wavelengths $\leq 500$ nm (AERONET daily mean). For a more detailed view, the vertically-resolved optical properties derived from the lidar were analyzed. They were compared to the lidar measurements from 16 September, which was before the start of the eruption and represents the typical aerosol conditions over Cabo Verde at this time of the year.

The lidar measurements for both days showed the presence of two distinct aerosol layers – the PBL and a lofted layer of Saharan dust. For the 24 September, HYSPLIT trajectory calculations and TRACE simulations indicated a distinct advection of air masses from La Palma in the PBL. Air masses of the lofted layer originated from the Saharan desert. With the lidar, a strong pollution in the PBL was revealed. It led to an unusual high particle extinction coefficient of $721 \pm 51$, $549 \pm 38$ and $178 \pm 13\,\text{Mm}^{-1}$ and an enhanced lidar ratio of $66.9 \pm 10.1$, $60.2 \pm 9.2$ and $30.8 \pm 8.7\,\text{sr}$ (mean values at 355, 532 and 1064 nm) in

contrast to $\leq 130\,\mathrm{Mm^{-1}}$ and $\leq 23.8\,\mathrm{sr}$ within the clean marine PBL on 16 September. Thus, on 24 September, the attenuation in the PBL was increased by a factor of 3–4 compared to the background conditions. The visibility significantly decreased during these days down to 6 km. According to the measured particle extinction coefficient, the AOD for the PBL was $0.58 \pm 0.03$, $0.43 \pm 0.02$ and $0.18 \pm 0.01$ at 355, 532 and 1064 nm, respectively. It accounts for 54-60 % of the total AOD in the case of 355 and 532 nm and for 32 % at 1064 nm. Compared to the AOD in the PBL of 0.08 (355 and 532 nm) during the clean marine PBL on 16 September 2021, we can conclude that the pollution on 24 September accounted for 81–86 % of the AOD in the PBL (AOD caused by pollution: $0.5 \pm 0.04$ and $0.35 \pm 0.03$ at 355 and 532 nm, respectively), i.e., only 14–19 % of the AOD in the PBL were caused by marine aerosol. Since the particle linear depolarization ratio in the PBL was close to 0 %, the presence of volcanic ash could be excluded. Instead, sulfate aerosol due to the volcanic eruption at La Palma seemed to be the dominating particle type in the low altitudes. This finding was furthermore supported by satellite measurements of Sentinel-5P, showing an advection of $SO_2$ towards Mindelo on 23 September, which was transformed to sulfate aerosol reaching Mindelo the day after.

In contrast, no indication for pure volcanic aerosol in the lofted layer could be found. The lidar ratio of $64.8 \pm 10.2$, $50.9 \pm 8.3$ and $61.8 \pm 8.6\,\mathrm{sr}$ (mean values at 355, 532 and 1064 nm) were slightly higher compared to $58.4 \pm 8.8$ and $47.3 \pm 7.2\,\mathrm{sr}$ (355, 532 nm) on 16 September 2021. Instead, the particle linear depolarization ratio of $20.8 \pm 2.0$, $25.0 \pm 1.0$ and $20.6 \pm 1.0$ % were lower than $24.5 \pm 2.0$, $28.1 \pm 1.0$ and $24.11.0$ % (355, 532, 1064 nm) observed on 16 September. However, Saharan dust as the major contributor can still be identified within this layer (SAL) but probably slightly contaminated with smoke, pollution and/or sulfate.

While observations of Saharan dust have already been captured during several campaigns (e.g., SAMUM; Ansmann et al., 2011a; Tesche et al., 2011), it was the first time that the optical properties of volcanic aerosol were observed at Cabo Verde with a multiwavelength-Raman-polarization lidar. Lidar observations of volcanic ash exist for distinct eruptions like Eyjafjallajökull (Ansmann et al., 2010; Groß et al., 2012) but lidar measurements of tropospheric volcanic sulfate aerosol are very rare, yet. Thus, it is important to enlarge the knowledge about the aerosol optical properties of volcanic sulfate, which is aimed by our study. As the observations were made in a usually clean marine PBL, the influence of mixing with other aerosol types is low. Besides this point, we show in our study that far-range transported volcanic aerosol can also effect air quality, indicated by sulfate mass concentrations of more than $100\,\mathrm{\mu g m^{-3}}$, and, thus, may have an impact on human health, even more than 1000 km away from the emission source. One additional benefit of this study is the first ever availability of measurements of the particle extinction coefficient and the lidar ratio at 1064 nm for volcanic sulfate. Having measurements at all three wavelengths is a major advantage with regard to lidar-based aerosol characterization and enlarges our data sets. The findings of this study can in turn be used to further improve the aerosol typing by multiwavelength-Raman-polarization lidars, as well as space-borne lidar missions as NASA's CALIPSO or ESA's Aeolus and EarthCARE, or assist in the development of new aerosol typing schemes. Besides this point, our findings will be helpful for studying the radiative effects of tropospheric volcanic aerosol, which are still not properly quantified and modeled. As the focus of the campaign at Cabo Verde was on the Aeolus validation, there is also the possibility for further research on the potential of Aeolus to capture the volcanic plume on its way to Cabo Verde, which is planned for future studies. Furthermore, a long-term study of the influence of the eruption of Cumbre Vieja on the atmosphere over Cabo Verde based on the ground-based lidar measurements at Mindelo is foreseen but first needs a

more robust cloud screening in the automatically derived lidar products. Finally, the observation of this event highlights the

necessity for ground-based lidar stations in remote areas. With respect to that, a permanent aerosol and cloud remote sensing station within the framework of ACTRIS has been set up in Mindelo.

*Data availability.* The Polly[XT] lidar data will be made available via ACTRIS services, but for now it is available at https://doi.org/10.5281/ zenodo.10650879 (Gebauer et al., 2024). Near-real-time measurement quicklooks can be found at https://polly.tropos.de/. AERONET data (station name "Mindelo_OSCM") was downloaded from https://aeronet.gsfc.nasa.gov/cgi-bin/draw_map_display_aod_v3?long1=-180&long2=

180&lat1=-89&lat2=90&multiplier=2&what_map=4&nachal=1&formatter=0&level=2&place_code=10 (AERONET, 2024), last access: 14 February 2024. HYSPLIT trajectories were calculated using the online tool on https://www.ready.noaa.gov/hypub-bin/trajtype.pl?runtype= archive with meteorological input data from GDAS1 (https://www.ready.noaa.gov/gdas1.php), last access: 23 January 2024. The data for the FLEXPART analysis was taken from https://doi.org/10.5065/D6M043C6, last access: 29 January 2024. The TROPOMI $SO_2$ plot was taken from https://so2.gsfc.nasa.gov/pix/daily/ixxxza/troploop5pca.php?yr=21&mo=09&dy=23&bn=cverde. The underlying data was downloaded

from the S5P-PAL data portal (https://data-portal.s5p-pal.com/) from July 2022 onward, and from BIRA-IASB distributions website (https: //distributions.aeronomie.be/) for older data, last access: 6 February 2024.

**Table 1.** Layer-mean values of the lidar-derived aerosol optical and microphysical properties for the PBL and the lofted layer on 24 September 2021, 04:38–05:57 UTC, and on 16 September 2021, 22:24–23:12 UTC. The values are given along with the standard deviation (parameter variability within the layer) for the extensive aerosol properties and the sulfate mass concentration and with the layer-mean errors (errors as described in Sec. 2.1) for the intensive aerosol properties. Geometric information of the aerosol layers is also provided (note that for the calculation of the extinction-related properties in the PBL the layer bottom height is 0.25 km due to the overlap configurations).

| | | Layer mean optical and microphysical properties | | | |
|---|---|---|---|---|---|
| | | PBL (nf, smoothing as in Fig. 5) | | lofted layer (ff, smoothing as in Figs. 3 and 4) | |
| | | 24 Sep | 16 Sep | 24 Sep | 16 Sep |
| | | 0.06–0.8 km | 0.06–0.6 km | 1.3–5.3 km | 1.4–4.4 km |
| | | sulfate | marine | Saharan dust | |
| **Extensive aerosol optical properties and microphysical properties** | | | | | |
| **Particle** | 355 nm | $10.9 \pm 0.6$ | $6.8 \pm 0.5$ | $2.0 \pm 0.9$ | $2.4 \pm 0.5$ |
| **backscatter** | 532 nm | $9.2 \pm 0.5$ | $5.6 \pm 0.4$ | $2.3 \pm 1.1$ | $2.9 \pm 0.6$ |
| **coefficient** | 1064 nm | $4.1 \pm 0.4$ | $3.6 \pm 0.3$ | $1.8 \pm 0.8$ | $1.8 \pm 0.3$ |
| $(Mm^{-1}sr^{-1})$ | 1064 nm (RR) | $6.1 \pm 0.3$ | - | $2.2 \pm 0.9$ | - |
| **Particle** | 355 nm | $721 \pm 51$ | $114 \pm 20$ | $120 \pm 64$ | $138 \pm 31$ |
| **extinction** | 532 nm | $549 \pm 38$ | $130 \pm 24$ | $114 \pm 65$ | $134 \pm 32$ |
| **coefficient** $(Mm^{-1})$ | 1064 nm | $178 \pm 13$ | - | $168 \pm 21$ | - |
| **Aerosol** | 355 nm | $0.58 \pm 0.03$ | $0.08 \pm 0.01$ | $0.38 \pm 0.25$ | $0.57 \pm 0.21$ |
| **optical** | 532 nm | $0.43 \pm 0.02$ | $0.08 \pm 0.01$ | $0.36 \pm 0.24$ | $0.57 \pm 0.20$ |
| **depth** | 1064 nm | $0.18 \pm 0.01$ | - | $0.39 \pm 0.16$ | - |
| **sulfate mass concentration** $(\mu g m^{-3})$ | | $133.1 \pm 20.3$ | - | - | - |
| **Intensive aerosol optical properties** | | | | | |
| **Lidar ratio** (sr) | 355 nm | $66.9 \pm 10.1$ | $17.3 \pm 2.8$ | $64.8 \pm 10.2$ | $58.4 \pm 8.8$ |
| | 532 nm | $60.2 \pm 9.2$ | $23.8 \pm 4.2$ | $50.9 \pm 8.3$ | $47.3 \pm 7.2$ |
| | 1064 nm | $30.8 \pm 8.7$ | - | $61.8 \pm 8.6$ | - |
| **Particle linear** | 355 nm | $0.3 \pm 2.0$ | $0.7 \pm 2.0$ | $20.8 \pm 2.0$ | $24.5 \pm 2.0$ |
| **depolarization** | 532 nm | $0.7 \pm 1.0$ | $1.1 \pm 1.0$ | $25.0 \pm 1.0$ | $28.1 \pm 1.0$ |
| **ratio** (%) | 1064 nm | $0.9 \pm 1.0$ | $1.0 \pm 1.0$ | $20.6 \pm 1.0$ | $24.1 \pm 1.0$ |
| **Ångström** | ext 355/532 nm | $0.68 \pm 0.07$ | $-0.32 \pm 0.29$ | $0.10 \pm 0.14$ | $0.06 \pm 0.08$ |
| **exponent** | ext 532/1064 nm | $1.53 \pm 0.26$ | - | $-0.06 \pm 0.53$ | - |
| | bsc 355/532 nm | $0.42 \pm 0.52$ | $0.46 \pm 0.52$ | $-0.43 \pm 0.52$ | $-0.47 \pm 0.52$ |
| | bsc 532/1064 nm | $1.13 \pm 0.31$ | $0.61 \pm 0.31$ | $0.40 \pm 0.31$ | $0.75 \pm 0.31$ |
| | bsc 532/1064 nm (RR) | $0.54 \pm 0.31$ | - | $0.13 \pm 0.31$ | - |

## Appendix A: AOD and lidar quicklooks

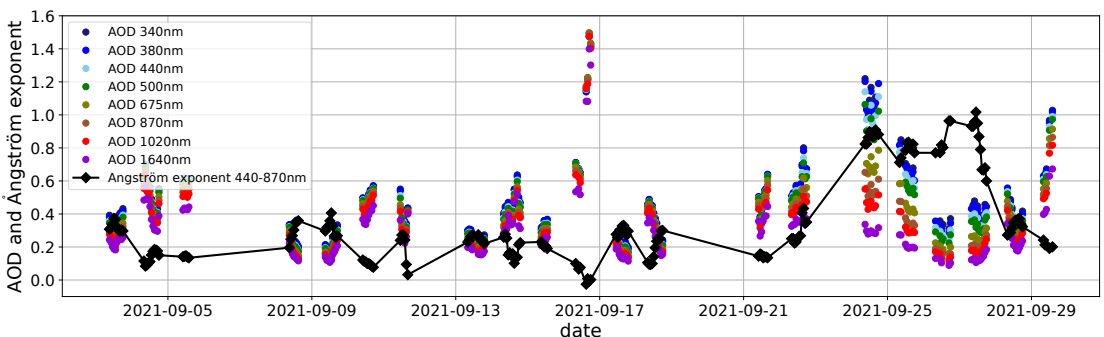

**Figure A1.** Same as Figure 1 but including the measurement of 16 September 2021, which was contaminated by a cirrus cloud.

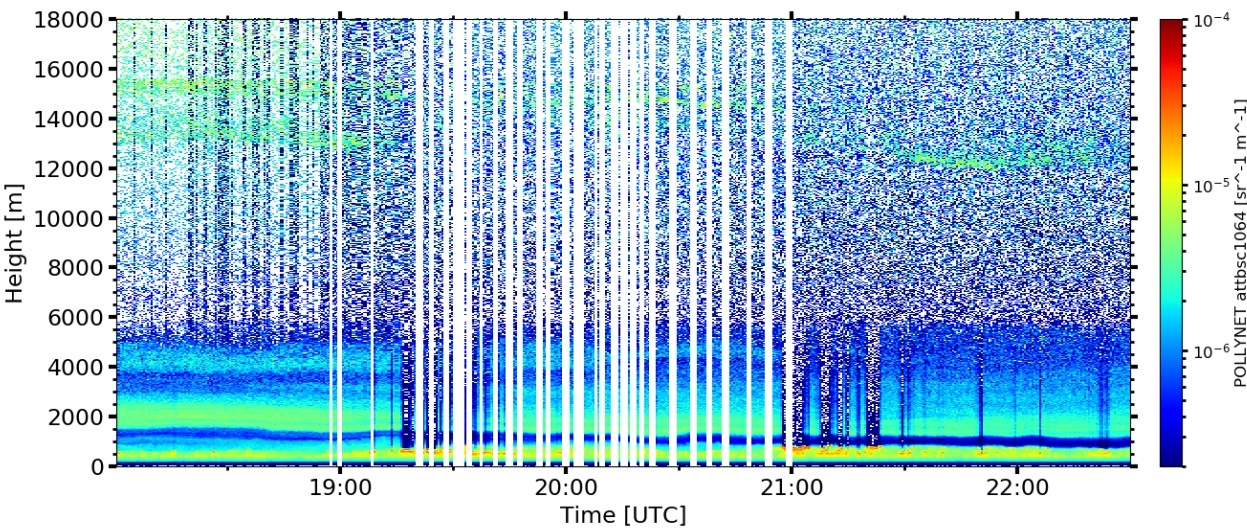

**Figure A2.** Temporal evolution of the calibrated attenuated backscatter coefficient at 1064 nm measured by Polly$^{XT}$ at Mindelo, Cabo Verde, during 16 September 2021, 18–22:30 UTC. In an altitude between 12 and 16 km the Cirrus cloud was located, which was not correctly screened out by the AERONET algorithm.

## Appendix B: Combined HYSPLIT and FIRMS maps

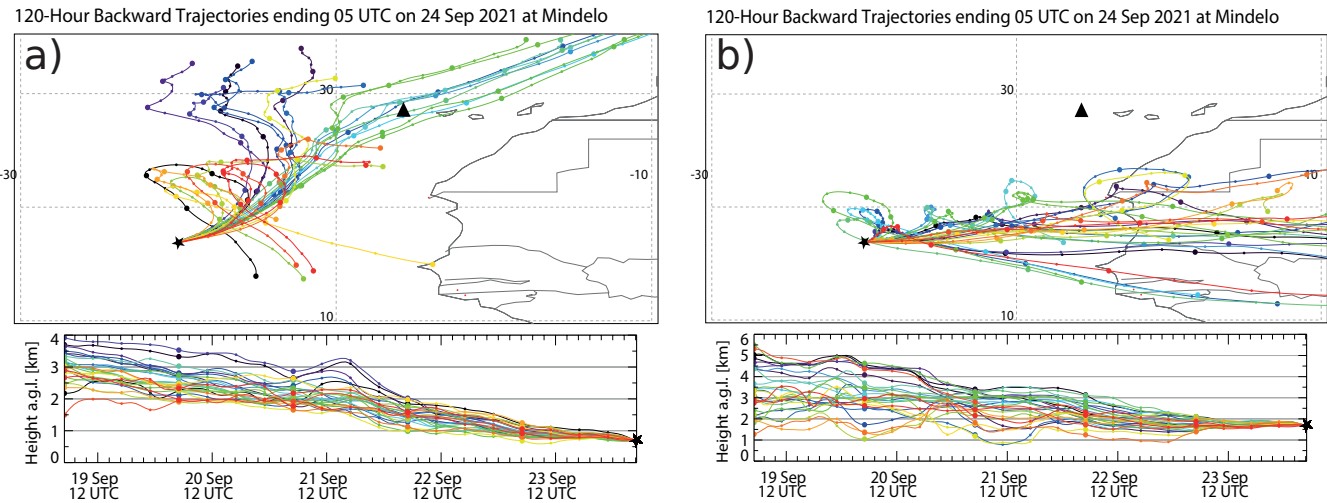

**Figure B1.** Same as Fig. 6 but with fire spot analysis from MODIS (FIRMS, firms.modaps.eosdis.nasa.gov) between 14 and 24 September 2021.

## Appendix C: TRACE simulations

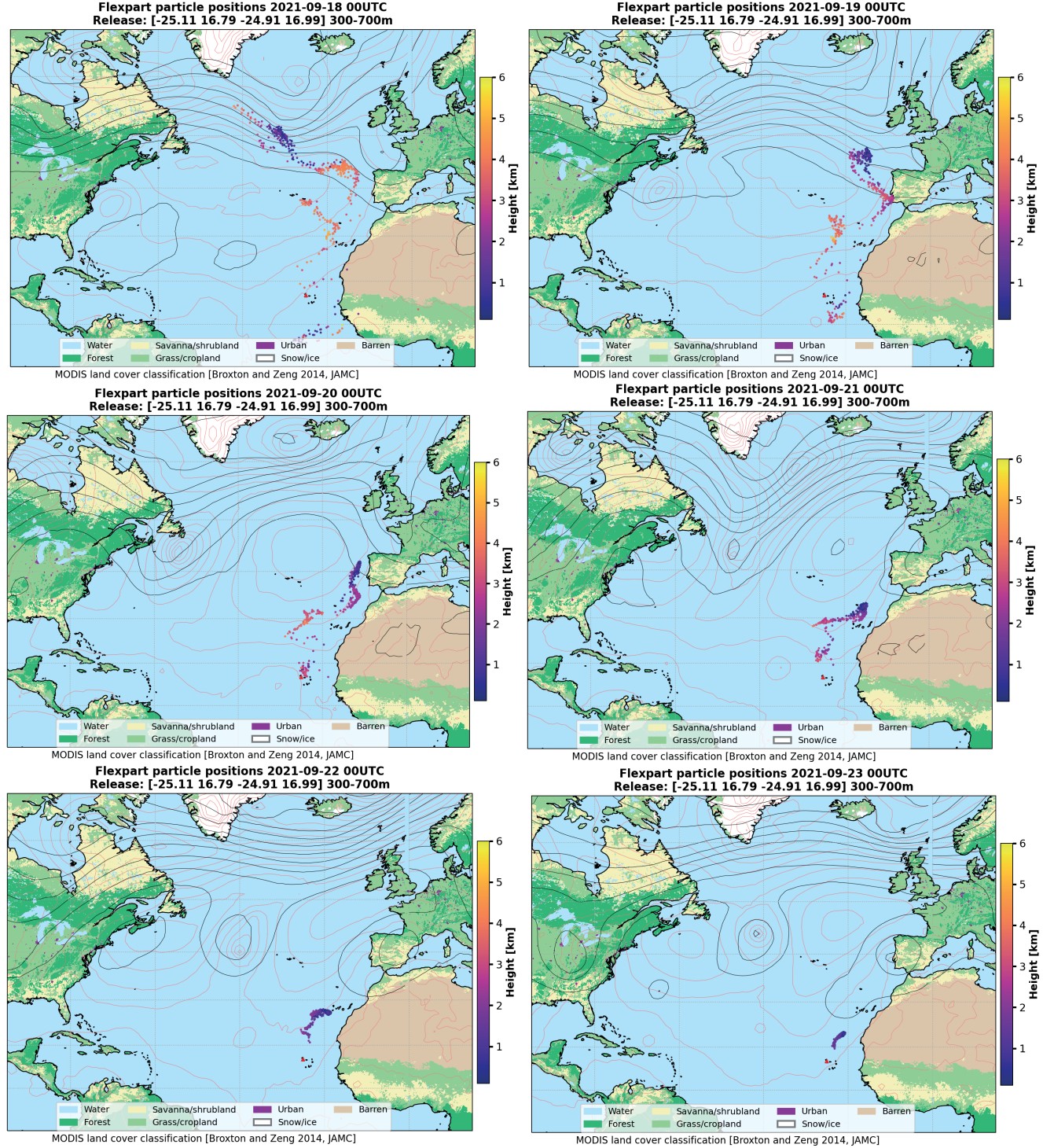

**Figure C1.** Selected TRACE simulations show the location of single air parcels (labelled as "particle positions") between 18 and 23 September 2021, each at 0 UTC, before they arrive at Mindelo (red triangle) at 500 m on 24 September 2021, 6 UTC. The color of the dots indicates their height above ground. MODIS land cover classification according to Broxton et al. (2014).

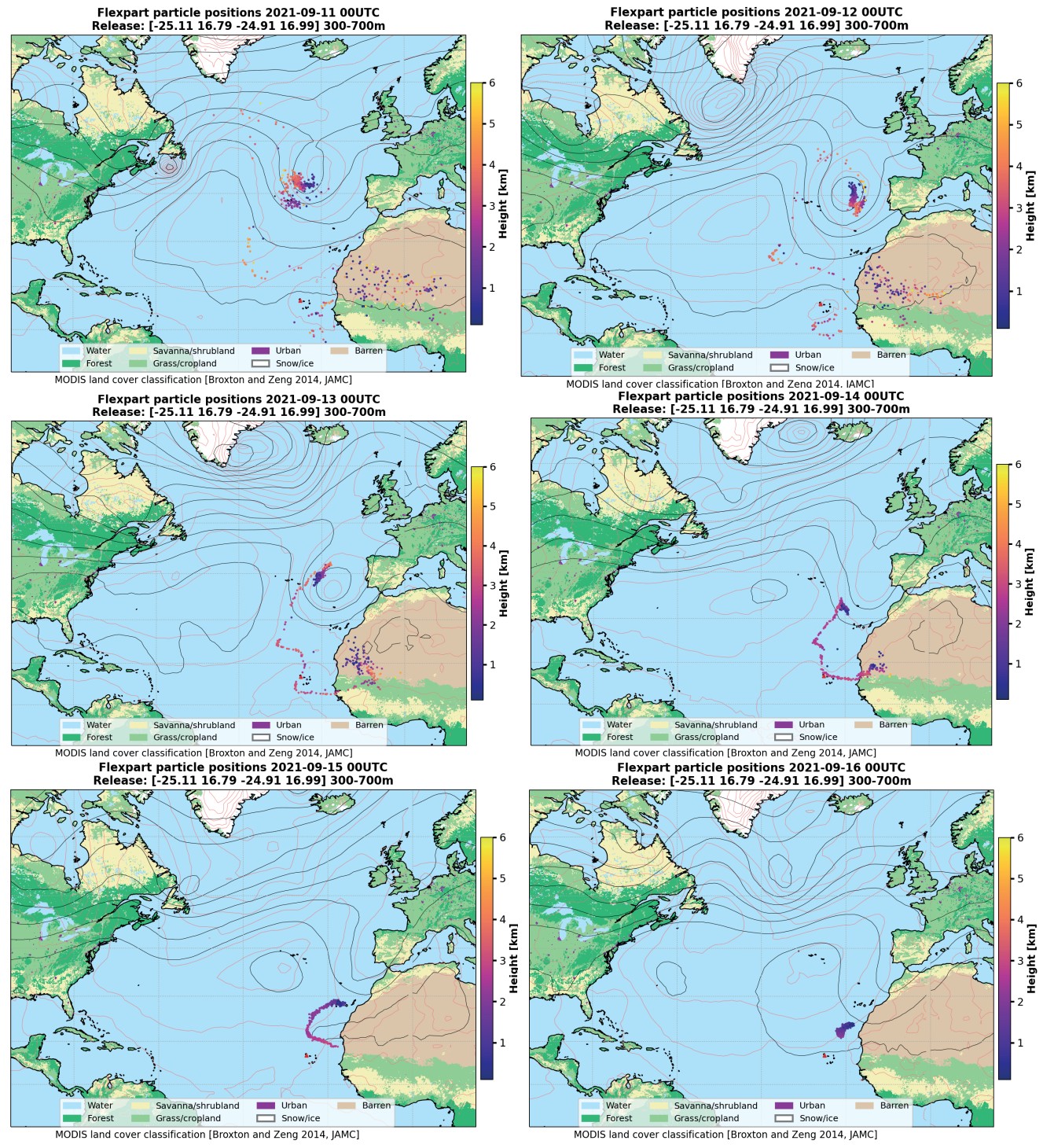

**Figure C2.** Same as Fig. C1 but for air parcels arriving on 16 September 2021, 0 UTC.

*Author contributions.* HG conceptualized the manuscript together with HB and AAF. MH provided the data of the lidar extinction measurements at 1064 nm. MR performed the TRACE simulations. AA contributed his expertise on lidar data analysis and volcanic aerosol. RE, DA, HB, and AS have been responsible for the deployment and operation of the ground-based instruments at Mindelo. CZ coordinates the scientific activities at OSCM, Cabo Verde. All coauthors were actively involved in the extended discussions and the elaboration of the final design of the manuscript.

*Competing interests.* The authors declare that they have no conflict of interest.

*Acknowledgements.* This research has been supported by the German Federal Ministry for Economic Affairs and Energy (BMWi) (grant no. 50EE1721C) and Horizon 2020 (grant nos. 654109 and 7395302). We furthermore acknowledge gratefully the team of OSCM for their support without which it would not have been possible to perform the observations. We also would like to thank ESA and the ASKOS/JATAC teams for the organization of the campaign and the support during the whole time!

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
