# Peer review of "Tropospheric sulfate from Cumbre Vieja (La Palma) observed over Cabo Verde contrasted to background conditions - lidar case study of aerosol extinction, backscatter, depolarization and lidar ratio profiles at 355, 532 and 1064 nm"

_EGUsphere, 2023_

## Author Comment (AC1)

**Response to reviewer 2:**

**General comments:**

The manuscript describes the measurements of a volcanic aerosol cloud from the Cumbre Vieja volcano eruption in Sept. 2021 that was transported 1500 km over the Atlantic Ocean from the island La Palma to the measurement site in Mindelo on the Cape Verde Islands. Measurements were performed with a multi-wavelength Raman and depolarisation lidar and a sun- and moon-photometer. Additionally, trajectory calculations are used to verify the origin of the measured airmasses in the boundary layer and in the free troposphere. The comparison of the measured values of the optical properties with the ones from a clean reference period a few days earlier and from lidar measurements of other campaigns indicate that the measured aerosol in the boundary layer is sulfate aerosol from the volcanic eruption.

As there are only few high quality multi-wavelength lidar measurements of aerosol stemming from volcanic eruption - especially spanning the wavelength range from 355 nm to 1064 nm, the retrieved aerosol optical properties add valuable data to the global database that can be used for aerosol characterization. The manuscript is very well organized and written.

*We greatly appreciate the review and detailed comments provided. As many comments were made with respect to the uncertainties of the lidar-derived optical products, we now have included a dedicated paragraph on this issue in the manuscript. You can find it in lines 141–150. Our responses to the specific comments are as follows.*

**Specific comments:**

Las Palmas is a municipality and city on Gran Canaria. The Cumbre Vieja volcano is located on the island called La Palma.

*Thank you for the correction! This point was also raised by RC1. We corrected the spelling of the island throughout the whole manuscript.*

The dual-field-of-view channels are mentioned in chapter 2.1 but unfortunately not used. Why? And if not used, why mentioning it?

*Thank you for your question. The dual-field-of-view (FOV) channels are mentioned in the manuscript as they are now part of the standard PollyXT setup. Since the dual-FOV polarization lidar technique was implemented only recently (Jimenez et al., 2020a, 2020b) it is not included in the instrument's overview paper (Engelmann et al., 2016). The technique is powerful, allowing the determination of microphysical liquid-water properties, which combined with the lidar-derived aerosol properties can be used for studying aerosol-cloud interactions. Since the scope of the paper is not aerosol-cloud-interactions we did not use the dual-FOV channels, nevertheless, their availability is worth mentioning. We have also included a relevant statement in lines 118–121 of the revised manuscript.*

The absolute scale of the "attenuated backscatter coefficient" shown in the fig. 2 need reference values at reference heights for each plot. This is even more necessary as measurements of two different days are compared (figs. 2c and 2d).

*It is worth to mention that here the calibrated attenuated backscatter coefficient is shown. Thus, it is a quantitative property which can be compared to each other. However, it is clear that different attenuation in lower altitudes leads to different values at higher altitudes even though aerosols properties there are similar. Thus, values are not comparable in terms of backscatter intensity at a given altitude. The aim of providing the plot is to show the vertical structure on the days of interest. We made this now clearer in lines 187–189 of the text.*

Figs. 3e and 4e: the x scales should be the same for an easier comparison of the values in the two plots.

*Thanks for the hint! We changed the x scales of Fig. 4e accordingly to the one of Fig. 3e. For the particle backscatter and extinction coefficient, we explicitly chose different x scales and prefer to leave it like that to provide as much detail as possible for the very different atmospheric conditions on these two days. Instead, we inserted the profiles from the Fig. 3 as reference in grey lines.*

Fig. 5b and 5c: the red 1064 nm (RR) line does not match the description in the figure caption. If an increased smoothing of 742.5m is used for the particle extinction coefficient and the lidar ratio at 1064nm, then there can be only one value in the considered height range between 0.25 km and 1.0 km, but there is a line of values between 0.75 km and 1.0 km.

*The vertical smoothing is performed by using a moving average filter, i.e., a vertical distance of 7.5 m between the data points is preserved. However, for an arbitrary smoothing length "s", each data point contains information of the height range from 0.5\*s below this point to 0.5\*s above it. Thus, with increasing s, the height, where the profile starts, increases, namely it starts always at 0.5\*s, but then it continues in steps of 7.5 m. For the 1064 RR products, with considering a smoothing length of 742.5 m, the data points of the extinction coefficient and lidar ratio in Figs. 5b and 5c (reaching now from 650m to 800m) contain information of the height range 278.5–1171.5 m. This limitation to this height range is done to exclude effects of the incomplete overlap (lower end) and noise (upper end). Please note the new height range of the 1064 extinctions products (650–800 m instead of 750–1000 m). We found a mistake in our plotting and had to adapt the range accordingly. This correction also led to minor changes with regard to the layer mean value, which we updated as well. Furthermore, we added an explanation concerning the smoothing to Sect. 2.1 (lines 141–150).*

The error bars in figs. 3 to 5 are not explained:

 1. What are the error bars showing? Should be mentioned in each figure caption.

*The error bars show the statistical error in case of the particle extinction coefficient and a relative error (minimized systematic error + statistical error) of 15 % for the particle backscatter coefficient. The errors of the lidar ratio and the Ångström exponent were calculated using the error propagation. For the particle linear depolarization ratio, constant absolute errors of 0.02 at 355 nm and 0.01 at 532 and 1064 nm are considered as described now in the manuscript in a dedicated section. We also clarified it now in the caption of Fig. 3 and refer to the new paragraph concerning the errors. As we avoided to repeat redundant information, which are the same in all three figures, we prefer to add the explanation of the error bars only to the caption of Fig. 3.*

2. Why does the error not change over large height ranges for e.g. in figs. 3c, 3d, and 3e, etc?

*This point should be clearer now with the better explanation of the errors in the dedicated section in lines 141–150. As the errors provided are relative ones, and the quantities itself in Fig. c,d,e are only slightly changing with height, it appears that the error bars are nearly constant.*

3. Figs. 3e and 4e should have the same x-scale for a better comparison.

*Done.*

4. The error bars in fig. 5b of the 355 nm and 532 nm are unrealistically small.

*We have added now a dedicated section (see lines 141–150) concerning the error calculation in the methodology part. Thus, the way we calculated the uncertainties should be clear. For the extinction, we obtain the uncertainty from the error of the linear fit made to get the deviation in the extinction formula. The resulting error is in the order of 10 Mm^-1 for the plotted height ranges and, thus, seems to be small compared to the large values we measured.*

5. In fig. 5c the error bar of the 1064 nm RR curve is much larger than the layer variability. What does it show?

*The error was calculated using the error propagation and includes the errors of the particle backscatter and extinction coefficient. With the added explanation of the errors in Sect. 2.1 (lines 141–150) it should be clearer now.*

The header of table 1 is a bit confusing:

*We rephrased the header now.*

1. In fig. 5b the values at 1 km height are already less than half of the mean values below. What does then the "edge effect" mean?

*"Edge effect" meant that we did not want to include the transition from the PBL to the lofted layer into the mean values for the PBL (and vice versa). Thus, we reflected that choosing 1 km as the upper boundary for the PBL was not appropriate and we changed it to 800 m. But we concluded, that "edge effect" is a misleading term and rephrased it accordingly.*

2. It is unclear what "standard deviation" means and how it is derived. Is it the uncertainty due to signal noise or the parameter variability over height?

*In this context, "standard deviation" means the statistical error due to the averaging over the layer, i.e., it is the parameter variability over height. As this quantity strongly depends on the vertical smoothing, we now decided to replace these values for the intensive optical properties with the layer mean of the given errors, which were used for the error bars in Figs. 3–5. We clarified this now also in the header of table 1.*

3. Is it possible that table 1 does not show any uncertainties?

*In the table of the preprint, the +- values did not include the uncertainties, which were used for the error bars in Figs. 3–5, but only the standard deviation (see answer above). We changed it now for the intensive optical properties, which do not show a significant vertical variability within a homogenous aerosol layer. For these parameters, the table now shows the layer mean error based on the uncertainties of the error bars. We clarified it in the table and, together with the new paragraph in Sect. 2.1 concerning errors/uncertainties, it should be clear, what is meant.*

 4. Why is there no 1064 nm AOD?

*Thanks for this hint. We now added it.*

 5. What do the +- values of the AOD mean? Standard deviation?

*The uncertainty values of the AOD describe the parameter variability. The AOD was calculated from the layer mean extinction coefficient while its uncertainty values were derived using the Gaussian error propagation with the standard deviation of the mean extinction as input. With the improved presentation of table 1, it now is clearer.*

The error bars / standard deviation values in the plots and table need a better discrimination and clearer description. This is especially necessary if these values should be used in other studies for aerosol typing.

*Thank you very much for the detailed feedback concerning the uncertainties! Of course, they are of particular importance! Thus, we added a new paragraph to Sect. 2.1 of the revised version of the manuscript, where all errors are explained in detail. Furthermore, we included a description of the used errors to the captions of the respective figures and the table.*

Furthermore, are systematic uncertainties considered?

*We calibrate our system according to ACTRIS/EARLINET standards. Thus, if we aware of any systematic error, we correct for it (e.g., depolarization calibration, polarization effects in the receiver unit). Remaining systematic errors are considered as described in the new paragraph concerning the uncertainties.*

Considering the uncertainties and variability, is the increase of the lidar ratio mentioned in lines 237ff really significant?

*You are right, it is not significant. We rephrased it accordingly.*

In line 293f it is stated about the AE values:

During the measurement period, the values decreased to 0.7 ± 0.1 and 1.7 ± 0.3, respectively, due to hygroscopic growth of the sulfate particles.

 => It is not clear, why the AE decreases. What is the difference between the aerosol parcels at the start and at the end of the "measurement period" - with respect to particle growth? Why do the latter grow more than the former?

*You are right, it is not clear, why the AE decreases. It was stated like this in the article of Navas-Guzman et al., 2013, but not further explained, how the hygroscopic growth may have*

*changed during the night. To answer your question, we can only speculate. One reason could be that the relative humidity increased during the night or that particles arriving at the end of the measurement period followed a different or longer trajectory so that they were exposed longer to high relative humidity. Also, a changing portion of sulfate particles and sulfuric acid droplets is possible. Either, more sulfate particles, which are the larger ones, are present at the end of the night, or there are more sulfuric acid droplets, which have a stronger hygroscopic change [Navas-Guzman et al., 2013]. Of course, coating and mixing with locally-produced aerosol cannot be excluded, leading to a change in the effective radius, too. As the aim of mentioning these findings was just to give reference values for aerosol optical properties of volcanic sulfate it is not meaningful to extend this discussion in the text. Thus, we clarified in the text the origin of this statement.*

**Some correction proposals:**

Line 28: diameter lower than => diameter smaller than

*Done.*

Line 63: ...which is in the range of 30 ±5% for pure dust ...  => at which wavelength?

*Done.*

Line 65: ...lower particle linear depolarization ratio ... => typical values?

*Done.*

Line 66:  ; John et al., 2011).  => (John et al., 2011).

*Done.*

Line 145: In addition, the add(horizontal) distribution of the volcanic plume was monitored.

*Done.*

Line 150:  A time series of the AOD => A time series of the CIMEL AOD at Mindelo

*We clarified that the columnar AOD from the sun photometer is meant and not the lidar-derived one.*

Line 151:   the hourly mean AOD was about 0.4.  =>  at which wavelength?

*Done.*

Line 154:    AODs, with values close to 1.0,  =>  at which wavelength?

*Done.*

Line 193:    than the once measured => ones

*Done.*

Line 195:   The lidar ratio at 1064nm is in line with dust observations at Leipzig, Germany  => are there no changes due to long range transport to be expected?

*The lidar ratio can be linked to the mineralogical composition of mineral dust, which depends mainly on the source region and does not change much during the transport. At least, previous observations of the lidar ratio at 355 and 532 nm pointed to this dependence (Veselovskii et al, 2020). The dust source region (Mauritania, Algeria or in more general terms Western Sahara) was similar for the observations at Leipzig in 2021 and the ones presented in the current manuscript. Therefore, we do expect a similar behavior for the dust lidar ratio. A comment about the similar source regions was added to the manuscript in lines 223–225.*

Line 207: In addition, vertical smoothing was reduced, which improves the accuracy of the near-field profiles => in which sense does the accuracy improve? The resolution is improved. But for which purpose?

*The resolution of the profiles in Fig. 5 was increased to reduce the overlap effect and to display the profiles down to lower altitudes than in Fig. 3 and 4, where a larger vertical smoothing was required to avoid too much noise in altitudes above 1 km. Furthermore, an improved vertical resolution allows us to better illustrate the vertical variability within the PBL. We clarified this now in the text. But you are right, the words accuracy, precision etc. should be used with more caution.*

Line 278: the measured quantities => Do you mean aerosol "quantities" or aerosol "optical properties"?

*You are right. "Aerosol optical properties" seems to be the more appropriate phrasing. We changed it now, also at equivalent positions in the text.*

Line 321: The intense pollution caused an unusually high AOD of more than 1.0 at different spectral bands. => change "different spectral bands" to "at the smaller wavelengths" - or so.

*Done.*

Line 351ff: " Having measurements at all three wavelengths allows us to get new insights in lidar-based aerosol typing and to enlarge our data sets. The findings of this study provide useful insights on the lidar-derived optical properties of volcanic aerosol."

=> This are diffuse statements. What are the new and useful insights?

*Thank you for this comment. You are right, these sentences were not clearly formulated and are redundant in the current state. We rephrased the paragraph to better point out the benefit of our study. The new and useful aspects of this study are that we provide aerosol optical properties measured with a multiwavelength-Raman-polarization lidar even at 1064 nm, which is a novel feature and rarely done so far. It was the first time that volcanic sulfate could be studied applying this technique. Our findings enlarge the existing data sets and can, thus, help to improve the lidar-based aerosol typing. Besides this aspect, they will also help to improve the quantification and modeling of the radiative effects of tropospheric volcanic aerosol.*

---

## Author Comment (AC2)

**Response to reviewer 1:**

**General Comments:**

In this manuscript, the contamination of long-range transported volcanic aerosol particles was captured in the local Planetary Boundary Layer (PBL) above the Ocean Science Center at Mindelo, Cabo Verde using ground-based lidar observations. More specifically, the authors use two case studies, one before (typical local PBL) and one during the 2021 eruption of the Cumbre Vieja volcano at La Palma island which is located at a distance of about 1500 km from the measurement site to showcase changes in the lidar optical properties and therefore demonstrate the importance of volcanic aerosol detection for health related purposes even further away from the volcanic activity. More specifically, the authors support on a single case study and they report the particle extinction coefficient, the particle extinction-to-backscatter ratio and the particle linear particle depolarization ratio at three wavelengths (355, 532 and 1064nm). The authors use auxiliary information to support that the changes inside the PBL are due to the volcanic activity alone using, for example, AERONET data, backward trajectories, fire presence using FIRMS (although not shown) and FLEXPART simulations. Given the need for accurate detection of volcanic particles in the atmosphere and the scarce lidar observations during volcanic eruptions the study has potential but in the current version it lacks scientific interest and clarity. Therefore, the manuscript may be considered for publication after major revisions.

*Thanks for the critical comment. We have now changed the focus of the manuscript a bit, so that the focus is more on the novel and unique optical properties rather than on the pollution event itself while preserving all main information. We also have changed the title so that it becomes clear that we focus on case studies. Furthermore, we completely revised the abstract so that the content of the manuscript should be much more obvious. With all these changes, we believe that the manuscript has significantly improved and also its scientific interest and clarity has become much clearer.*

**Specific comments:**

The title is misleading. In the manuscript, two case studies using a PollyXT lidar are shown. One before the eruption at La Palma island and a second one during the volcanic activity. Then, the authors use the two case studies to discuss the changes in the lidar optical properties over the measurement site. Therefore, I suggest choosing a more suitable title including the word case study. Please note and correct throughout the manuscript: The name of the island is La Palma and should not be confused with Las Palmas which is the capital of Gan Canaria.

*Thank you for making us realize that the title was misleading. We have now changed our title to "Tropospheric sulfate from Cumbre Vieja (La Palma) observed over Cabo Verde contrasted to background conditions – lidar case study of aerosol extinction, backscatter, depolarization and lidar ratio profiles at 355, 532 and 1064 nm", which reflects better the contents of the manuscript. Additionally, we have also corrected the spelling of "La Palma" throughout the manuscript. Many thanks for this hint!*

The abstract should be short, clear, and summarize the findings of the study. As written, the abstract is misleading and lacks scientific importance. It is misleading because, the authors mention the full duration of the volcanic activity, but they do not mention that the findings

rise from one individual case study. It also lacks scientific importance because although it is mentioned that a special version of a PollyXT system was used which allowed the calculation of the particle extinction coefficient, the particle extinction-to-backscatter ratio and the particle linear particle depolarization ratio at three wavelengths (355, 532 and 1064nm), there is no mentioning of the 1064nm wavelength which is the added value compared to the standard high-power lidars which operate at 355 and/or 532nm. In fact, there is no comprehensive summary of the findings per wavelength per aerosol optical property. More specifically, the lidar ratio and linear particle depolarization ratio which are of great importance in aerosol classification are absent from the abstract.

*Thank you for your critical but constructive comment. We have restructured and shortened the abstract to point out that we performed a case study and not a long-term analysis and highlighted the availability of the measurements at 1064 nm. Furthermore, we added the results, especially for the lidar ratio and the depolarization ratio, at the particular wavelengths.*

The authors support the changes in the PBL optical properties during the volcanic eruption to be caused by the presence of sulphate aerosols and they exclude the presence of other aerosol sources using FLEXPART and backward trajectories together with fire location from FIRMS. It would be beneficial to include a FIRMS figure and I was also wondering whether there are in situ observations at Mindelo site to check the presence of black/organic carbon. This will solidify your conclusions regarding the higher lidar ratio observed during the volcanic activity and its origin.

*Thank you for your recommendations! We added a FIRMS figure in combination with backward trajectories to the appendix B of the manuscript. We also agree that in-situ observations would be helpful for the argumentation and checked the availability of in-situ data (e.g., in the Global Atmosphere Watch World data center https://ebas-data.nilu.no/) but, unfortunately, no measurement data of black or organic carbon at Mindelo is publicly available for that time period.*

Then, I really missed a long-term report of the lidar optical properties including the full period of the volcanic activity. The authors advertise in quite a few points in the manuscript that the lidar has captured the full volcanic activity. So, why did you choose to focus on one case study only and not include the full dataset? Furthermore, it will be of great importance to go a step further and estimate the mass concentration of the long-range transported sulphate aerosols.

*Thank you for this comment! In this manuscript, we focus on the case study analysis and the respective optical properties. Given the revised title and abstract it should become clearer now. But we are aware that a long-term study of the full period of volcanic activity would be of large interest. Continuous lidar measurements were performed during that time, but a complete quality control data set including successful and reliable cloud screening as a prerequisite for a high-quality aerosol optical data set is not yet available. That is why we, for now, analyzed single days within this period. However, a long-term study based on automatically retrieved lidar optical properties may be subject of a future publication as we added in the outlook. For now, we reworked the misleading paragraphs in the manuscript. Furthermore, we now added a profile of the sulfate mass concentration to Fig. 5 and a corresponding paragraph in lines 347–353 of the revised manuscript to highlight the potential of such lidar measurements for studies of air quality.*

Overall, in its current form the manuscript is a report of a single case study in which the optical properties are the result of marine and sulphate aerosol mixture of unknown contribution with limited added value to the scientific community and at places highly speculative. The presentation and usage of English should be also substantially improved.

*With the new title and the updated abstract, the focus of the manuscript should be clearer. Several new aspects are considered with the case study: It is the first time, optical properties at 1064 nm for sulfate aerosol are measured. We can clearly disentangle the effect of the sulfate from marine by comparing to background conditions and, thus, show the impact of the volcanic eruption at Mindelo on this specific day. We could show that no ash was transported from the volcano towards Mindelo. Furthermore, we reworked the manuscript linguistically. With all these changes, we believe that the manuscript is of high value for the scientific community.*

L122-124: Can the authors comment on the stability of the calibration for this wavelength? What is the expected error in the optical products? The references in this sentence point to another lidar system and not PollyXT.  What is the error estimation for this system? Furthermore, the 1064nm depolarization capability is also a new feature. What is the uncertainty of the particle depolarization ratio for this system?

*The calculation of the extinction coefficient at 1064 nm via the rotational Raman method follows the methodology described in Haarig et al., 2016. The spectral cross-talk calibration methodology uses a liquid cloud as it was introduced in Haarig et al., 2022 and is referenced in the manuscript but briefly described here for your convenience: the strong backscatter signal at the cloud base is used to iteratively determine the spectral cross-talk correction factor. The elastic signal (1064 nm) multiplied with the spectral cross-talk correction factor is subtracted from the rotational Raman signal (1058 nm) so that the particle induced strong backscatter signal at cloud base (elastic backscattering process) is not contaminating the rotational Raman signal any more. The spectral cross-talk correction factor changes only when the neutral density filters are changed in one of the two channels (either 1064 or 1058 nm). As there was no change in neutral density filters between 24 September and 4 October 2021, the spectral cross-talk correction factor of 6.7e-4 +/- 0.3e-4 is valid for the whole period.*

*The calibration of the depolarization ratio at 1064 nm and the estimation of its uncertainties followed the same approach as the calibration at 355 and 532 nm (Engelmann et al., 2016). The Delta 90° calibration (Freudenthaler et al., 2009) with a linear polarizer after the pinhole was applied.*

*We also included a dedicated section describing the uncertainty estimation for all optical products now in the manuscript – see lines 141–150 of the revised manuscript. Following this section, the error for the particle depolarization ratio is about 0.01 at 1064 nm.*

L178-179: Is there a reference to support the statement that the measurement on the 16[th] of September represent the typical situation over the location?

*The lidar-derived optical properties have been already studied, quite intensively, in the framework of ASKOS and L2A+ (ESA funded projects). A comprehensive overview of the PollyXT lidar measurements conducted during the ASKOS intensive measurement periods is shown in Fig. S1. The attenuated backscatter coefficient at 1064 nm (Fig. 1a, 1c, 1e) in combination with*

*the volume depolarization ratio at 532 nm (Fig. 1b, 1d, 1f) reveal the typical aerosol conditions above Mindelo, which are a clean marine boundary layer (MBL; non-depolarizing spherical particles), with a dust aerosol layer (depolarizing non-spherical particles) on top of that.*

[Figure]

**Figure S1: Overview of the lidar attenuated backscatter coefficient at 1064 nm (left column) and volume depolarization ratio at 532 nm (right column) as retrieved from the PollyXT lidar during the ASKOS operations in September 2021 (a, b), June 2022 (c, d), and September 2022 (e, f).**

*These results have been published (for now) in project deliverables (e.g., https://l2a.space.noa.gr/backend/assets/e7a1b125-e2b2-4489-a205-720bd4f8077a?download, last access: 11 January 2024). Additionally, the data are available in native format via the ESA Validation Data Center (EVDC) under DOI: 10.60621/jatac.campaign.2021.2022.caboverde (2023). We have now included the relevant citations in lines 208–211 of the revised manuscript.*

Figures 3-5: What the error bars refer to? Do they include the systematic errors or just the variability caused by the time averaging?

*The error bars show the statistical error in case of the particle extinction coefficient and a relative error (minimized systematic error + statistical error) of 15 % for the particle backscatter coefficient. The errors of the lidar ratio and the Ångström exponent were calculated using the error propagation. For the particle linear depolarization ratio, constant absolute errors of 0.02 at 355 nm and 0.01 at 532 and 1064 nm are considered as described now in the manuscript. Thus, minimized systematic errors are included as well as statistical errors. We added a detailed explanation of the uncertainties in Sect. 2.1 (lines 141–150) and clarified the meaning of the error bars in the figure caption.*

**Technical corrections:**

Lines 67-69: Please give a reference.

*Done, please see line 70 in the revised manuscript.*

Lines 75-86: Note this publication about Cumbre Vieja volcanic eruption:

Bedoya-Velásquez, A.E.; Hoyos-Restrepo, M.; Barreto, A.; García, R.D.; Romero-Campos, P.M.; García, O.; Ramos, R.; Roininen, R.; Toledano, C.; Sicard, M.; et al. Estimation of the Mass Concentration of Volcanic Ash Using Ceilometers: Study of Fresh and Transported Plumes from La Palma Volcano. Remote Sens. 2022, 14, 5680. https://doi.org/10.3390/rs14225680

*Thank you for pointing to this interesting publication that was missing in our review! We included it now!*

Line 104: Please check that the coordinates of the measurement site are correct. I think it should be W, not E.

*We corrected the wrong coordinate of the measurement site as well as of the volcano. Thank you for notifying!*

Figures 2-5: Use the same thickness for all lines.

*The different line thickness was chosen to satisfy the requirements for the color blindness tests but thanks to your comment we noticed that this solution may be confusing. Thus, we standardized the line thicknesses now.*

Figure 3: Be consistent on the heights at which the error bars appear. For example, Figs. 5a, b and c all have the error bar in different locations.

*We chose the different heights on purpose to ensure a better visibility of the error bars but we also acknowledge your comment as, of course, choosing consistent heights ensures a better comparability. Thus, we changed it according to your recommendation.*

Figure 4d: ref. lines are missing from the legend. For example, the dotted orange line which most probably refers to the bae532/1064 (RR) on the 16[th] of September.

*We added the missing symbols and labels in the legend.*

Figure C1: Include the 16[th] of September 2021 as it is one of the case studies. The figure could also be a bit more zoomed to the region of interest.

*Thanks for this suggestion. We have included it now accordingly.*

---

## Author Response (AR2)

**Response concerning the technical corrections**

**Response to the reviewer**

Thank you for raising both important points!

We tackled both and clarified it in the manuscript.

First topic:

We added another sentence for a better explanation in lines 148 – 149. Now, it reads like this: "For the particle extinction coefficient, the statistical error is calculated from the error of the linear fit of the derivative without considering systematic uncertainties. This linear fit considers as much data points as the smoothing length ($s$) and is applied every 7.5 m." It should now be clear that the linear fit is used to obtain the derivation of the lidar signal.

Second topic:

We agree with you that the use of percent for the depolarization ratio should be avoided. We plan to use the legacy diction for future publications but do not want to introduce too many changes at this stage of the manuscript preparation. Thus, we prefer to leave it like it is in this manuscript but will keep it in mind for future publications. But we changed the statement concerning the change of properties and added the term "relative" so that it should be now clear to understand: "Considering the wavelength dependence of the particle linear depolarization ratio, a relative decrease of 18 % from 532 towards 1064 nm was observed. Similar findings were made at Leipzig, Germany, and Morocco during SAMUM (relative decrease by 13 – 31 % […])." (See lines 231 – 232 in the new manuscript.)

**Response to the editor**

Thank you for carefully reading and for your hints! We implemented all technical corrections you proposed.

**P4, L103:** ASKOS is not an abbreviation but a Greek word originating from mythology. We added its meaning and stated that it is "just" a campaign name.

**P5, L139:** In this case, $s$ is not a unit but the symbol of the smoothing length. For clarification, we added the multiplication sign.

**P8, Figure 3 caption:** The typical half spaces between number and unit exist already. We double-checked it.

**P15, L317:** You wrote: 'Add "volcanic" after "eruption"'. Probably, you mean „before" instead of "after". We added it there.